mathematical modelling/bioengineering/ computer modelling and simulation

actigraphy, sleep, chronic insomnia, multi-night recordings, dynamical features, machine learning

**Author for correspondence:**
M. Angelova
e-mail: maia.a@deakin.edu.au

# A machine learning model for multi-night actigraphic detection of chronic insomnia: development and validation of a pre-screening tool

S. Kusmakar[1], C. Karmakar[1], Y. Zhu[1], S. Shelyag[1], S. P. A. Drummond[2], J. G. Ellis[3] and M. Angelova[1]

[1]School of Information Technology, Deakin University, Geelong, Victoria 3125, Australia
[2]Turner Institute for Brain and Mental Health, School of Psychological Sciences, Monash University, Melbourne, Australia
[3]Department of Psychology, Northumbria University, Newcastle upon Tyne, UK

MA, 0000-0002-0931-0916

We propose a novel machine learning-based method for analysing multi-night actigraphy signals to objectively classify and differentiate nocturnal awakenings in individuals with chronic insomnia (CI) and their cohabiting healthy partners. We analysed nocturnal actigraphy signals from 40 cohabiting couples with one partner seeking treatment for insomnia. We extracted 12 time-domain dynamic and nonlinear features from the actigraphy signals to classify nocturnal awakenings in healthy individuals and those with CI. These features were then used to train two machine learning classifiers, random forest (RF) and support vector machine (SVM). An optimization algorithm that incorporated the predicted quality of each night for each individual was used to classify individuals into CI or healthy sleepers. Using the proposed actigraphic signal analysis technique, coupled with a rigorous leave-one-out validation approach, we achieved a classification accuracy of 80% (sensitivity: 76%, specificity: 82%) in classifying CI individuals and their healthy bed partners. The RF classifier (accuracy: 80%) showed a better performance than SVM (accuracy: 75%). Our approach to analysing the multi-night nocturnal actigraphy recordings provides a new method for screening individuals with CI, using wrist-actigraphy devices, facilitating home monitoring.

# Statement of significance

Nocturnal awakenings affect the quality of sleep which in turn can have implications on an individual's health. Assessment of nocturnal awakenings during sleep, using in-laboratory polysomnography, is a resource-intensive procedure and also brings the individual out of their normal home environment. Furthermore, patient self-report and sleep diaries can introduce a subjective bias. Automated ambulatory monitoring techniques can facilitate an objective assessment and also allow for longitudinal testing over several nights. This study proposes a machine learning framework that provides an automated analysis of actigraphy recordings to objectively discriminate between individuals with insomnia and healthy sleepers.

# 1. Introduction

Sleep is an essentially biological process with important roles in regeneration, conservation of energy and survival [1]. Sleep also seems important for vital functions such as metabolic, cardiovascular, emotional regulation, neural development and the elimination of cellular toxins [2–5]. Sleep, both in terms of quality and quantity, is affected by sleep disorders like narcolepsy [6], sleep apnoea [7] and insomnia [8]. Of these, insomnia is one of the most common sleep disorders [9]. The DSM-5 defines insomnia based on dissatisfaction with sleep associated with one (or more) of the following symptoms: sleep initiation difficulties, difficulties maintaining sleep: characterized by frequent awakenings or problems returning to sleep after awakenings, and/or early morning awakenings. Insomnia can be acute or chronic depending on its duration, with the transition point being three months. Chronic insomnia (CI) has been associated with daytime cognitive deficits, exhaustion, depression, anxiety, stress, reduced quality-of-life, increased use of health services, and greater use of hypnotic and stimulants for sleep disorders [10–12]. While the incidence of CI is high [13], a considerable number of individuals with CI do not seek medical attention and may not even realize their sleep is unhealthy. When individuals do seek medical attention, many primary care providers are unsure of how to diagnose insomnia. These problems suggest a potential role for easily implementable methods designed to identify potential insomnia that can then trigger the individual or their healthcare provider to seek specialized assessment and diagnosis.

Actigraphy is a non-invasive method for monitoring human rest/activity patterns. Actigraphs can infer sleep characteristics from the motor activity and are widely used to measure sleep in the home environment, including individuals with insomnia [14–16]. However, actigraphy is currently not able to reliably distinguish those with insomnia from healthy sleepers, and thus cannot currently play the screening role suggested above [14,16]. Recently, we have reported a machine learning-based model for the detection and classification of acute insomnia using wrist-actigraphy [17]. We found that multi-night actigraphy recordings can efficiently capture acute insomnia signatures and distinguish those with acute insomnia from healthy sleepers. In addition, actigraphy is less obtrusive and more convenient for continuous monitoring over days or weeks and provides an economical and non-invasive means of assessing sleep [18,19]. These advantages contribute to the actigraph's feasibility of estimating and assessing the sleep characteristics in the ambulatory setting. Motivated by these advantages, we propose in this study a wrist-actigraphy-based automated tool for pre-screening individuals in their natural environment and providing an automated detection of CI.

Multi-night recordings are seldom used for the detection of CI and have not been used before for building an automated model for such detection. Most studies rely on manual labelling of nights (good/bad) using sleep diaries; however, the use of manual labelling introduces a user dependency and such models are not suitable for automated pre-screening. Therefore, we propose a fully automated framework for detecting individuals with CI that automates the procedure of labelling in a data-driven manner. Our study aims to assess the efficacy of multi-night actigraphy recordings to build and validate a machine learning model for automated pre-screening of CI in adults.

There are several potential advantages of an automated pre-screening tool that can identify individuals with likely insomnia and in need of a more thorough assessment. Despite the relative simplicity of actigraphic measurements, the efficacy of different actigraphy-derived statistical and sleep parameters remains to be investigated. Furthermore, deriving sleep parameters without any additional information from patient sleep diaries is a challenging problem in actigraphy research [20,21]. Such a scenario would be particularly beneficial for populations that cannot easily use sleep diaries (e.g. those with physical or intellectual disabilities), as it is well known that reports by others

(e.g. carers) are not well correlated with the individuals' reports [17]. Therefore, the use of actigraphs as an inexpensive and non-invasive secondary measure of insomnia without the use of sleep diaries warrants investigating novel and reliable methods to extract wake pattern indicators. In this study, we have used features extracted directly from actigraphy time-series signal: statistical and nonlinear features in addition to a subset of sleep parameters to enhance the performance of the developed model. Furthermore, since machine learning models usually provide better performance using a larger number of informative features, we have not limited the model to use only sleep-based parameters and added other features. As a result, in this study, we use advanced signal analysis techniques to extract dynamical and nonlinear time-domain features from actigraphy signals. We further investigate the use of actigraphy as a stand-alone method for a data-driven automated detection of CI in an adult population. This, in turn, will help in establishing the utility of wearable actigraphy-based devices for the prospective assessment of insomnia patients in their natural environment. The major contributions of the study are as follows:

— A machine learning model for evaluation of CI using multi-night actigraphy signals.
— An automated wearable solution with potential for pre-screening of individuals with CI.

## 2. Methods

### 2.1. Preliminary work

In our previous study [17], we conducted a successful validation for the detection of acute insomnia using dynamic time-domain features extracted from multi-night actigraphy signals. The accuracy of the developed model was 84%, with a sensitivity of 76% and specificity of 92% in correctly classifying individuals with acute insomnia. From the analysis reported in [17], we found that our model was able to deduce the signatures of acute insomnia from the wrist-worn actigraphy devices with seven nights of data. The algorithm reported here further extends our work to CI subjects with a focus on developing an automated actigraphy-based insomnia pre-screening tool. In this study, we apply the algorithm on a bigger dataset comprising continuous long-term multi-night actigraphy recordings from 80 individuals to derive robust signatures to accurately detect and classify individuals with CI.

Figure 1 demonstrates the workflow of the proposed actigraphy-based CI detection model. The following sections will describe the workflow in detail.

### 2.2. Subjects

Our analysis uses actigraphy time-series data. In this study, we used the actigraphy data recorded from cohabiting couples with one subject seeking treatment for their insomnia. Subjects were recruited through online advertisements, radio, print media and via referrals from a psychologist, sleep clinics or general practitioner (GP) in Melbourne, Australia. The subjects were recruited as part of a larger ongoing clinical trial (Project REST; Australian New Zealand Clinical Trials Registry Registration: ACTRN12616000586415), to study partner-associated behavioural interventions in insomnia [15,22]. The study was approved by the human research ethics committee of Monash University.

### 2.3. Protocol

The inclusion criteria for the couples were set as: (i) age should be above 18, (ii) partners have shared a bed for over one month, (iii) there was no information of continuing domestic abuse in the relationship, and (iv) were fluent in English. The criteria for insomnia disorder were established according to the Duke Structured Interview for Sleep Disorders (DSISD) in individuals experiencing insomnia [23]. The exclusion criteria included: (i) the individual experiencing insomnia had any other unmanaged sleep disorders, assessed through DSISD and a polysomnography (PSG) diagnosis, (ii) had undergone behavioural treatment therapy in the past month, (iii) had reported a diagnosis of manic episodes or unmanaged psychosis, (iv) has a substance abuse or alcohol disorder in the last month, (v) completed shift work during the last month, (vi) transmeridian trips over more than one-time zone (included after a waiting period of one week), (vii) in the first trimester of pregnancy and (viii) below the consenting age. Exclusion criteria for the healthy partner ('non-patient partner') included: (i) a history of uncontrolled psychosis or manic episodes and (ii) a history of substance abuse or alcohol disorder. The couple was not enrolled if either partner was ineligible. In our previous studies, we have found

**Figure 1.** The workflow of the proposed CI detection model.

**Table 1.** The variation in the number of chronic and healthy pairs and the total nights of recording available for analysis based on the number of nights of recorded actigraphy data.

| | nights | | | | | | |
|---|---|---|---|---|---|---|---|
| subject | 1 | 2 | 3 | 4 | 5 | 6 | 7 |
| chronic/healthy pairs[a] | 40 | 33 | 32 | 31 | 25 | 23 | 19 |
| total nights[b] | 80 | 132 | 192 | 248 | 250 | 276 | 266 |

[a]The number of subjects corresponding to different nights of recording.
[b]The total number of night's recordings available for building the model.

that actigraphy-measured sleep was extremely similar in the partners of those with insomnia and the individuals in the couples where both partners were normal sleepers [15,24]. However, although insignificantly, one bed partner can indeed affect the sleep of the other bed partner and, therefore, if our algorithms can separate the two groups in this more challenging situation, they were likely to do even better when individuals are truly independent. Participating in the current procedure involved an assessment session with the researchers, a night of PSG to rule out the possibility of any occult sleep disorders, and seven nights of in-home sleep tracking within one month of the assessment session (0–34 day afterwards; $M = 9.5$, s.d. $= 9.5$ days). Full details of the protocol for the data collection are given in Mellor *et al.* [22].

In total, this study comprised 80 adults; 40 healthy controls and 40 subjects with CI. The two groups were matched on age because of the partner status. The current study used wrist-actigraphy in recruited subjects and the data were collected for one week using actiwatch model Respironics Actiwatch Spectrum Pro and raw data processed with Actiware software (Respironics, Bend, OR, USA). Movement counts were sampled in 60 s epochs. All recruited subjects wore the devices at all times during day and night. All subjects were free to move and were not prohibited from doing any activities of daily living. The data collection for each participant included five working days and one weekend. The participants were not restricted on which specific day of the week to start the measurements. However, not all subjects continued for the entire one-week period. Table 1 shows the variation in the number of subjects and total nights of recording. The data for this study were obtained from the first week of assessment and the baseline evaluation. Figure 2 shows the raw actigraphy data for seven nights of recording from an individual with CI and the cohabiting healthy partner. The demographics of the recruited individuals are shown in table 2.

## 2.4. Data pre-processing

The raw actigraphy data were pre-processed and prepared for feature extraction. Only the subjects' nocturnal actigraphy signals, corresponding to the time in bed (TIB), were used for feature extraction. Actigraphy data were extracted without the use of sleep diaries as a guide to determine lights out

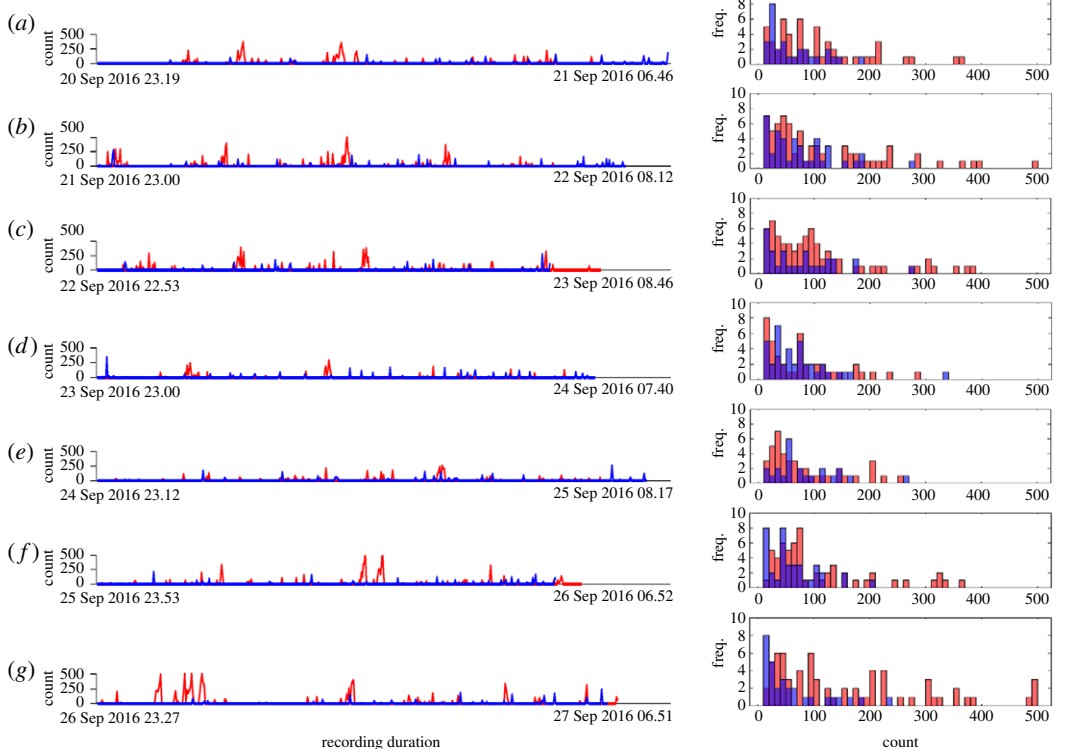

**Figure 2.** An example figure showing the raw nocturnal actigraphy data for seven nights (*a–g*) of recording. Shown are the recordings for individuals with CI (in red) and their healthy cohabiting partners (in blue). Also shown are the distributions of the raw signal; CI (red bars) and healthy (blue bars). It can be seen that subjects with CI have higher activity in comparison to their healthy cohabiting partners.

**Table 2.** The total number, male to female ratio and mean age of CI and healthy individuals recruited in the study.

| total | CI[a] | healthy |
|---|---|---|
| subjects | 40 | 40 |
| male : female | 13 : 27 | 25 : 15 |
| mean age | 48.03 | 47.2 |

[a]CI, chronic insomnia individuals.

and lights on. Rather, the start and end of the night were determined automatically by the actigraphy software. While this may introduce small errors, especially in sleep latency (SL), our stated goal was to develop an automated detection algorithm that requires no input from the subject. Moreover, it has been shown that actigraphy-driven rest intervals show a high correlation (intraclass correlations range 0.864–0.995) with manual scores of bedtimes and rise time [25]. Therefore, in this study, we have used ActiWatch-generated rest intervals as TIB.

Actigraphy data are a measure of the number of times the acceleration of an activity exceeds a reference value (summed over 1 min epochs). The wake sensitivity limit could be set by the clinician or analyst as high (20 counts per epoch), medium (40 counts per epoch) low (80 counts per epoch). There are no clear instructions, however, on how to use these limits [26]. Additionally, the use of the wake sensitivity limit serves as a low-pass filter, which eliminates subtle variations from the nocturnal actigraphy signal. Thus, features extracted after applying different filtering levels are not going to be the same and can contain diverse details. As a result, signal filtering will further enrich the feature set derived from the nocturnal actigraphy signal that the machine learning model will use to distinguish subjects with CI from healthy sleep partners. In this study, we employed four different wake limits (intensity filters (InF)) of zero, low, medium and high activity counts of 0, 20, 40 and 80, respectively,

**Table 3.** The list of the features extracted from the nocturnal actigraphy signal.

| measure | index | feature name |
|---|---|---|
| statistical features | 1 | mean (arithmetic) |
| | 2 | s.d. (standard deviation) |
| Poincaré plot features | 3 | SD1 |
| | 4 | SD2 |
| | 5 | ratio (SD1/SD2) |
| | 6 | CCM (complex correlation measure) |
| entropy | 7 | SampEn (sample entropy) |
| Sleep parameters | 8 | TST (total sleep time) |
| | 9 | SL (sleep latency) |
| | 10 | WASO (wake time after sleep onset) |
| | 11 | SWR (sleep–wake ratio) |
| | 12 | SE (sleep efficiency) |

on the nocturnal actigraphy signal. The use of similar filtering has been shown previously by Alexandre Domingues & Sanches [27]. The filtering sets the actigraphy signal counts to zero if their values are less than or equal to the InF value. The filtered nocturnal actigraphy signals are then processed for feature extraction. As we have four InF levels ($InF_1 = 0$, $InF_2 = 20$, $InF_3 = 40$ and $InF_4 = 80$), the filtering procedure results in four different sets of features from one night of raw actigraphy signal.

## 2.5. Feature extraction

We extracted linear and nonlinear features from the nocturnal actigraphy signal such as mean, standard deviation (s.d.), Poincaré plot features (SD1, SD2, ratio, area, CCM) and entropy-derived measure (sample entropy *SampEn*). In addition, features such as total sleep time (TST), SL, wake after sleep onset (WASO), sleep–wake ratio (SWR) and sleep efficiency (SE) were also extracted from the nocturnal actigraphy signal. Thus, a total of 12 different features were extracted from each nocturnal actigraphy signal (table 3). As the study employs four different InF levels on the raw nocturnal actigraphy signals, it generates four unique signals and a feature set comprising 48 ($12 \times 4 = 48$) features corresponding to each night of recorded actigraphy data.

(1) *Statistical features:* The numerical mean and s.d. of the nocturnal actigraphy signal were calculated to determine the average amplitude of the nocturnal actigraphy signal and the sum of variance. Both the average amplitude and variance were found to be greater for individuals with CI than their healthy partners.

(2) *Poincaré plot features:* The Poincaré plot method quantifies the self-similarity between time-series signals. It is a well-known technique that has been extensively used in the analysis and quantification of biomedical signals like heart rate variability. It is derived from the nonlinear dynamical theory to represent the variations in heart rate [28]. The Poincaré plot has also been employed in the analysis of activity data from physical sensors like accelerometers for quantifying seizure-like activity [29,30]. The Poincaré plot is a two-dimensional point cloud in which one point in the plot represents a successive sample of a time sequence. SD1 (short-term variability), SD2 (long-term variability) and their ratio (SD1/SD2) are the typical descriptors or parameters used to quantify the plot [31]. In addition, Poincaré can also be used to measure the dynamical properties of the time-series sequence [32].

(3) *Sample entropy (SampEn):* Sample entropy (SampEn) quantifies the irregularity of a non-stationary time-series signal. It has been widely used in analysing physiological signals. SampEn is an estimate of the conditional likelihood of two segments matching at a length of $m + 1$ if they match at $m$, where the tolerance parameter $r$ determines the match [33]. Self-matches are omitted in SampEn, when measuring the percentage of vectors that are with the tolerance $r$. In this study, the SampEn of the actigraphy signal was used as a feature to quantify the wake epochs.

(4) *Actigraphy-derived sleep parameters:* From the nocturnal actigraphy signal the following sleep parameters TST (total sleep time), WASO (wake after sleep onset), SWR (sleep–wake ratio), SL (sleep latency) and SE (sleep efficiency) were calculated.

TST is the total duration of all sleep epochs (activity count is zero) starting from the SO (sleep onset) time. The SO time is defined as the start of the first 10 min epoch of continuous inactivity (activity count is zero) in the nocturnal actigraphy signal.

SL is defined as the duration between sleep onset SO and the start time of nocturnal recording.

WASO is the period, measured in minutes, by summing all wake epochs in the nocturnal actigraphy signal following the SO time.

The ratio of TST and WASO is called SWR, as shown in the below equation

$$SWR = \frac{TST}{WASO}. \tag{2.1}$$

The ratio of TST and rest interval, TIB (the rest interval generated by the actiwatch), known as SE or % sleep [34], and is given by the below equation

$$SE = \frac{TST}{TIB}. \tag{2.2}$$

## 2.6. Pattern classification model

In this study, a novel two-layered automated model was employed for detecting individuals with CI (figure 1). The first layer of the model works by predicting the night labels for all the subjects. In order to avoid the use of the sleep diaries, we labelled each night of data from healthy sleepers as a 'good night of sleep' and that of the CI individuals as a 'bad night of sleep', respectively. The produced night labels were then used to train a supervised classifier to predict the night labels for the test subject during the cross-validation process. In this study, we employed a leave-one-out cross-validation technique to predict the night labels of each left-out subject. Let $N$ represent the total number of subjects, then during each iteration of the cross-validation, the classifier sequentially leaves out one subject and predicts the night labels based on the model trained from $N-1$ subjects. The cross-validation procedure ensures that a label is predicted corresponding to all nights in every subject. Therefore, the first layer predicts the night labels for the entire dataset. Two supervised models of machine learning, namely support vector machine (SVM) and random forests (RF), were used to construct the prediction model. Given a relatively small amount of data, not sufficient for deep learning algorithms to function properly, and the requirement for basic explainability for the models we create, we use these traditional methods [35].

The frequency of bad nights (predicted at layer 1 by the ML model) is used as a criterion for individual classification in the second layer of the model. A threshold value for the number of bad nights to be classified as an individual with CI was determined for each subject. This threshold was optimized in a leave-one-out fashion, where the predicted night labels of $N-1$ subjects were used to determine the threshold, which was then used to classify the left-out subject as belonging to the CI or healthy groups. Therefore, the predicted labels of all subjects were aggregated after $N$ iterations, and used for performance analysis.

(1) *Support Vector Machine:* SVMs are state-of-the-art binary state classifiers widely employed in a variety of research domains [36,37]. The SVM classifier maps each instance from the feature space to a high-dimensional space such that the data can be separated linearly using a decision boundary learned in the form of a hyperplane. This mapping is done using kernels and provides SVMs with the ability to learn nonlinear functions and decision boundaries. For simplicity, we employed a linear kernel with BoxConstraint = 1, in this study and all predictors are standardized before training the classifier to ensure feature commensurability. All other sensitive hyperparameters such as $\eta$ and bias are optimized by the 10-fold cross-validation.

(2) *Random forest:* RF belongs to one of the most popular ensemble learning classification methods [38]. The method independently builds a large set of decision trees using the training dataset, so that the nodes of each decision tree are a randomly selected subset of features from the dataset.

A decision tree is a data analytics approach that affords exploration of the presence of complicated interaction patterns within the data by creating binary individual segmentations into subgroups.

Subgroup membership in a decision tree is derived from the response to a set of measured/predictive variables in the training set. A decision tree is so-called because in a tree-like structure the predictive variables are interpreted, beginning from a single node and branching into multiple nodes indicating possible choices [39]. The decision, when predicting a variable, follows the path from the root node down to a leaf node along the tree. Decision trees are efficient and find application in a variety of research domains [40].

RF builds a host of decision trees as a forest using the entire training dataset. Given the test data, RF determines the final classification result by using the predicted classes of the individual trees. This ensemble approach of multiple individual classifiers greatly boosts the classification performance. The RF parameters used in the study were: 100 trees and the maximum tree depth was set to the size of the training set. The square root of the number of features is selected at random for each split. We have applied a 10-fold cross-validation method to select the optimal model.

## 2.7. Performance metrics

Once all classification results have been obtained (from the two layers of the model for detecting individuals with CI), three standard measures (sensitivity, specificity and accuracy) of classifier performance evaluation were applied to evaluate each classifier's efficiency [41]. The performance measures are defined as follows:

(i) $\text{Sensitivity} = \dfrac{\text{TP}}{\text{TP} + \text{FN}} \times 100$, i.e. the true positive rate in per cent.

(ii) $\text{Specificity} = \dfrac{\text{TN}}{\text{TN} + \text{FP}} \times 100$, i.e. the true negative rate in per cent.

(iii) $\text{Accuracy} = \dfrac{\text{TP} + \text{TN}}{\text{TP} + \text{FP} + \text{TN} + \text{FN}} \times 100$.

For the subject-level classification results, sensitivity refers to the proportion of CI subjects classified correctly, and specificity refers to the proportion of healthy partners that are correctly classified. The model accuracy refers to the rate of correctly classifying individuals with CI from their healthy partners.

# 3. Results

This research uses seven nights of nocturnal actigraphy data for developing a machine learning-based model for automated detection and differentiation of individuals with CI from their healthy partners. The participation of healthy bed partners instead of independent healthy controls showed that the proposed model is very effective, sensitive and accurate even with noisy data.

## 3.1. Feature values and patterns

Table 4 summarizes the mean and standard deviation values of features for healthy sleepers and individuals with CI. The feature values are constantly higher in individuals with CI in comparison to their cohabiting healthy partners, except for nonlinear features (ratio, CCM, SampEn) and TST that show the opposite trend (table 4 and figure 3). Statistical features like mean and standard deviation of the nocturnal actigraphy signal showed a statistically significant ($p < 0.05$) difference between individuals with CI and their healthy partners. The Poincaré plot-derived features like SD1 and SD2 show a statistically significant difference between CI and healthy individuals, whereas ratio and CCM have a lower differentiating capacity at zero intensity filtering level and improve with increasing intensity filtering levels (figure 4 and table 4). The sample entropy *SampEn* as a feature showed a statistically significant improvement in differentiating individuals with CI and healthy individuals as *InF* value was increased (figures 3 and 4 and table 4). Sleep parameters like TST, SE and WASO have been previously employed for assessing healthy individuals using actigraphy [42]. All sleep parameters calculated in the study except SL tend to a statistically significant difference between feature values for individuals with CI as the intensity filter value increased (table 4). This was also evident by the increase in the area under the receiver operating characteristic curve (AUC) value, as shown in figure 4. The pattern of values for all the different intensity filters used in this analysis remains the same or similar. Among all features, mean and SD1 show the highest differential capacity among the two groups of subjects as shown by the AUC values (figure 4). The AUC values for the individual features (table 4) indicate that there is not a single predictor with sufficient capacity to

**Table 4.** The mean $\pm$ s.d. (standard deviation) values of features for the healthy sleepers and CI for (a) $InF_1 = 0$; (b) $InF_2 = 20$; (c) $InF_3 = 40$ and (d) $InF_4 = 80$.

| feature | healthy | CI | p-value[a] | AUC[b] |
|---|---|---|---|---|
| (a) $InF_1 = 0$ | | | | |
| mean | 69.12 ± 24.20 | 93.72 ± 37.19 | 0 | 0.72 |
| s.d. | 88.79 ± 36.56 | 121.76 ± 56.40 | 0 | 0.68 |
| SD1 | 36.05 ± 12.93 | 47.99 ± 17.65 | 0 | 0.7 |
| SD2 | 50.30 ± 26.94 | 70.23 ± 41.00 | 0 | 0.68 |
| ratio | 0.78 ± 0.18 | 0.75 ± 0.16 | 0.055 | 0.57 |
| CCM | 0.27 ± 0.05 | 0.27 ± 0.04 | 0.473 | 0.53 |
| SampEn | 0.18 ± 0.04 | 0.18 ± 0.06 | 0.284 | 0.54 |
| TST | 413.83 ± 118.98 | 392.50 ± 80.24 | 0.318 | 0.54 |
| WASO | 79.11 ± 37.35 | 78.21 ± 31.35 | 0.775 | 0.51 |
| SWR | 0.19 ± 0.07 | 0.21 ± 0.13 | 0.584 | 0.52 |
| SL | 15.98 ± 30.78 | 12.44 ± 28.11 | 0.688 | 0.51 |
| SE | 81.77 ± 6.86 | 81.49 ± 7.48 | 0.943 | 0.5 |
| (b) $InF_2 = 20$ | | | | |
| mean | 107.18 ± 31.79 | 130.89 ± 48.47 | 0 | 0.66 |
| s.d. | 96.87 ± 45.10 | 129.92 ± 64.78 | 0 | 0.66 |
| SD1 | 36.21 ± 12.95 | 48.18 ± 17.66 | 0 | 0.7 |
| SD2 | 50.27 ± 27.02 | 70.22 ± 41.06 | 0 | 0.68 |
| ratio | 0.79 ± 0.18 | 0.75 ± 0.16 | 0.048 | 0.57 |
| CCM | 0.27 ± 0.05 | 0.27 ± 0.04 | 0.461 | 0.53 |
| SampEn | 0.15 ± 0.04 | 0.16 ± 0.05 | 0.211 | 0.54 |
| TST | 454.35 ± 125.72 | 421.97 ± 80.43 | 0.057 | 0.57 |
| WASO | 50.38 ± 26.42 | 55.78 ± 24.97 | 0.008 | 0.59 |
| SWR | 0.11 ± 0.04 | 0.14 ± 0.09 | 0 | 0.63 |
| SL | 4.20 ± 11.40 | 5.40 ± 11.70 | 0.14 | 0.54 |
| SE | 89.67 ± 3.64 | 87.62 ± 5.42 | 0 | 0.63 |
| (c) $InF_3 = 40$ | | | | |
| mean | 133.44 ± 37.19 | 155.75 ± 52.45 | 0 | 0.63 |
| s.d. | 99.51 ± 49.15 | 133.76 ± 68.96 | 0 | 0.65 |
| SD1 | 36.22 ± 13.09 | 48.35 ± 17.75 | 0 | 0.7 |
| SD2 | 50.00 ± 27.17 | 70.00 ± 41.24 | 0 | 0.67 |
| ratio | 0.80 ± 0.18 | 0.76 ± 0.16 | 0.066 | 0.57 |
| CCM | 0.27 ± 0.05 | 0.27 ± 0.05 | 0.648 | 0.52 |
| SampEn | 0.12 ± 0.03 | 0.14 ± 0.04 | 0 | 0.65 |
| TST | 468.47 ± 130.27 | 435.20 ± 81.53 | 0.058 | 0.57 |
| WASO | 37.05 ± 19.98 | 44.15 ± 22.35 | 0 | 0.63 |
| SWR | 0.08 ± 0.03 | 0.10 ± 0.07 | 0 | 0.7 |
| SL | 3.40 ± 10.97 | 3.80 ± 7.93 | 0.439 | 0.52 |
| SE | 92.40 ± 3.15 | 90.36 ± 4.48 | 0 | 0.68 |

(Continued.)

**Table 4.** (Continued.)

| feature | healthy | CI | p-value[a] | AUC[b] |
|---|---|---|---|---|
| **(d) InF$_4$ = 80** | | | | |
| mean | 181.30 ± 47.91 | 209.11 ± 62.25 | 0 | 0.64 |
| s.d. | 101.54 ± 55.76 | 139.99 ± 78.01 | 0 | 0.65 |
| SD1 | 35.56 ± 13.56 | 48.00 ± 18.50 | 0 | 0.7 |
| SD2 | 48.64 ± 27.71 | 68.82 ± 41.89 | 0 | 0.67 |
| ratio | 0.81 ± 0.19 | 0.77 ± 0.17 | 0.08 | 0.56 |
| CCM | 0.27 ± 0.05 | 0.27 ± 0.05 | 0.606 | 0.52 |
| SampEn | 0.07 ± 0.03 | 0.09 ± 0.03 | 0 | 0.66 |
| TST | 484.51 ± 134.84 | 452.70 ± 84.58 | 0.092 | 0.56 |
| WASO | 22.67 ± 15.06 | 28.17 ± 19.05 | 0 | 0.63 |
| SWR | 0.05 ± 0.02 | 0.06 ± 0.06 | 0 | 0.66 |
| SL | 1.74 ± 2.57 | 2.29 ± 4.22 | 0.536 | 0.51 |
| SE | 95.52 ± 2.07 | 93.95 ± 3.82 | 0 | 0.67 |

[a]The non-parametric Mann–Whitney $U$-test was used to calculate the $p$-value and the significance was determined for $p < 0.05$. $p < 0.05$ indicates that the feature values are different for individuals with CI and their healthy partners.

[b]AUC is the area under the receiver operator characteristics curve (ROC). An AUC = 1.0 indicates complete separability, whereas an AUC = 0.5 indicates that the model is not able to distinguish between healthy and CI class.

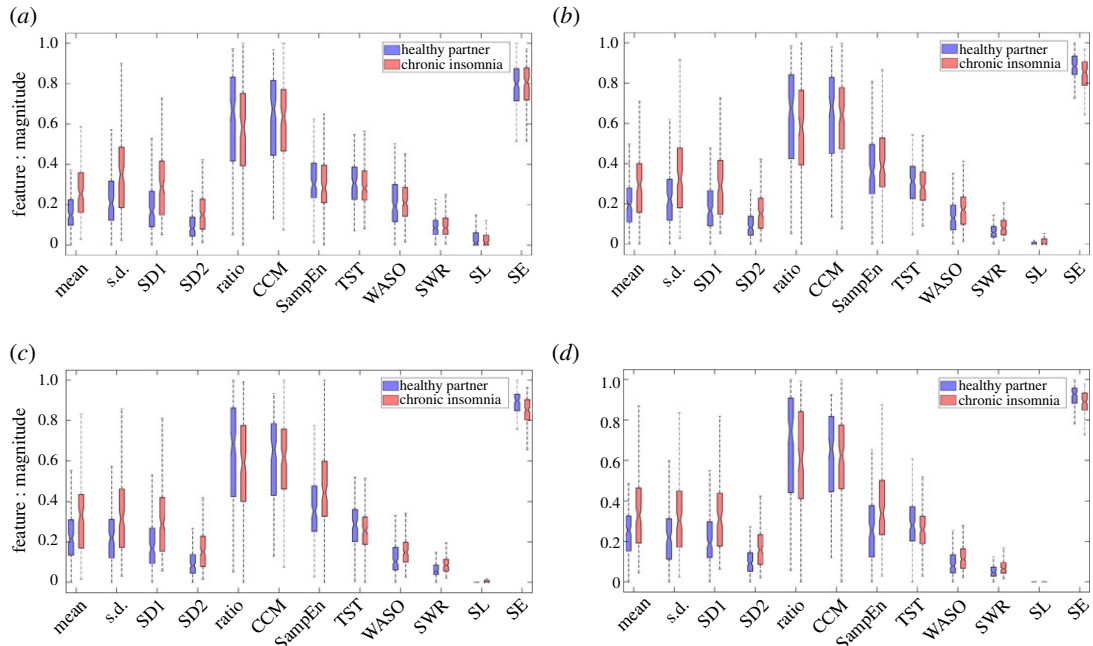

**Figure 3.** The box and whisker plot of the features corresponding to the four intensity filtering levels: ($a$) InF$_1$ = 0, ($b$) InF$_2$ = 20, ($c$) InF$_3$ = 40, ($d$) InF$_4$ = 80 on the nocturnal actigraphy data for individuals with CI and their cohabiting healthy partners. For visualization, the raw feature set is normalized in the range [0, 1].

differentiate between CI and healthy sleep groups and justify the use of all of the 48 features listed in table 4a–d for developing the machine learning models.

## 3.2. Classification of chronic insomnia

This study proposes an automated machine learning method for the detection of individuals with CI using seven nights of actigraphy data recorded using a wrist-actigraphy device. A two-layered model is developed, where the first layer predicts the quality of night sleep and classifies it into good or bad

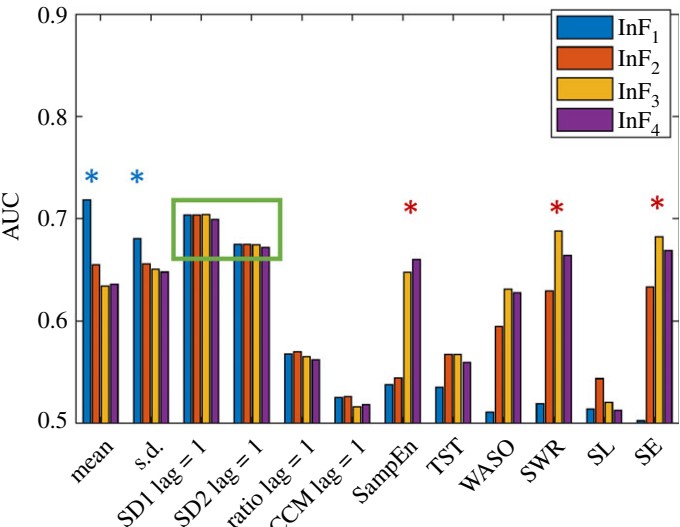

**Figure 4.** The feature AUC values shown for different intensity filtering (InF) levels (InF$_1$ = 0, InF$_2$ = 20, InF$_3$ = 40 and InF$_4$ = 80). The blue asterisk symbol indicates the features that have the highest AUC at lower intensity level (InF$_1$ = 0), while red asterisk indicates the features that perform better at high-intensity filtering level (InF$_3$ = 40 and InF$_4$ = 80). Some features are unaffected by intensity filtering as highlighted by the green rectangular box.

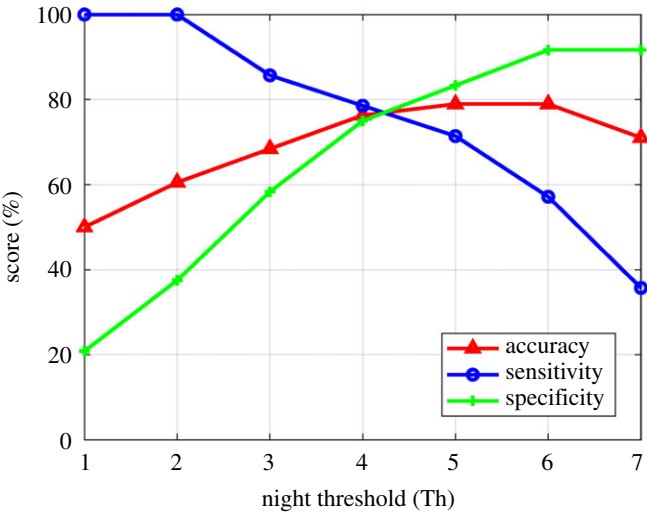

**Figure 5.** The model determined optimal bad night threshold (Th): performance measures (accuracy, sensitivity, specificity) corresponding to varying night thresholds (Th), defined as the number of bad nights shown here for the model developed using the RF classifier.

night sleep. A second optimizer layer uses the predicted quality of sleep (good/bad) to assign the individual to the CI or healthy group (figure 1). A leave-one-out validation approach was used to determine the optimal night threshold Th, ensuring that the test subject is not a part of the learning process. To understand the night threshold (Th) use, let us assume a model generated an optimal night threshold of Th = $x$. If for an individual, the Layer 1 of the model predicted greater than or equal to $x$ nights of the total monitoring nights as bad nights, he/she would be classified into the CI group. Figure 5 shows the optimal night threshold Th value for the RF classifier. The learned thresholds for both RF and SVM classifier are four nights, respectively, for all iterations of the learning. Table 5 shows the overall model classification performance using the optimal night threshold.

The RF classifier shows a higher overall classification accuracy of 80% in comparison to the SVM classifier that showed a classification accuracy of 75% in classifying individuals with CI from their healthy cohabiting partners. In addition, the model performance corresponding to both the classifiers (RF and SVM) increases with the length (total analysed nights) of the nocturnal actigraphy recordings (figure 6).

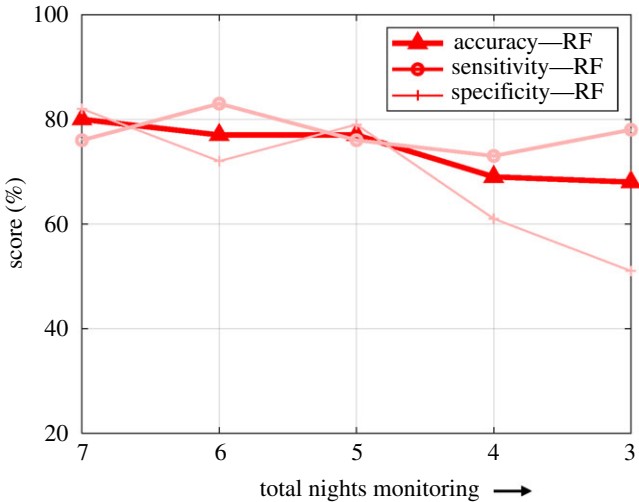

**Figure 6.** The subject-level performance of the model using RF classifier. The classifiers show the highest classification (CI and healthy cohabiting partners) performance for seven nights of recording and the model performance decreases as the number of total analysed nights are reduced.

**Table 5.** The classification performance (accuracy, sensitivity and specificity) of the best fit model for RF and SVM classifier for (a) night level and (b) subject-level classification. The optimal learned night threshold for classifying an individual as CI was Th = 4 for both RF and SVM classifiers. (a) Performance of the model at layer 1: the night-level classification performance for classification of every individual night into good or bad using RF and SVM classifiers. (b) Performance of the model at layer 2: subject-level classification performance on incorporating optimal night threshold (Th).

| algorithm | accuracy | sensitivity | specificity |
|---|---|---|---|
| (a) performance of the model at layer 1 | | | |
| RF | 73% | 68% | 77% |
| SVM | 68% | 69% | 67% |
| (b) performance of the model at layer 2 | | | |
| RF and Th = 4 | 80% | 76% | 82% |
| SVM and Th = 4 | 75% | 79% | 73% |

The optimal classification performance is achieved for seven nights of recordings and the model classification performance drops to a random guess for a recording duration below five nights. The classification performance of the RF and SVM classifiers were also verified by plotting its ROC curve and comparing the AUC of the curves (figure 7). As evident from the curve (figure 7), the AUC of RF was higher than the SVM classifier, which indicated better classification performance.

## 3.3. Effect of randomization

We hypothesized that a healthy sleeper will have more good nights sleep, per week, than a subject with CI. To test our hypothesis, we recalculated performances of the proposed model for individual nights and subjects after random labelling of the nights. After randomly labelling all the nights for the subjects, the model showed an overall classification accuracy of 56% (sensitivity: 23%, specificity: 71%) for RF, and 48% (sensitivity: 24%, specificity: 64%) for the SVM classifier, respectively. In comparison, the proposed approach where the night labels are first predicted using our data-driven approach shows a much-improved performance (accuracy: 80%, sensitivity: 76%, specificity: 82%, using RF).

This clearly shows that the labelling of each night as good/bad captures the inherent patterns in the data. In addition, our hypothesis is influenced by the clinical observations that the quality of sleep in healthy individuals is much higher than in the individual with CI. Therefore, blind labelling of a night of a healthy sleeper as a good night and of a subject with CI as a bad night, although not exactly accurate, still retains the pattern.

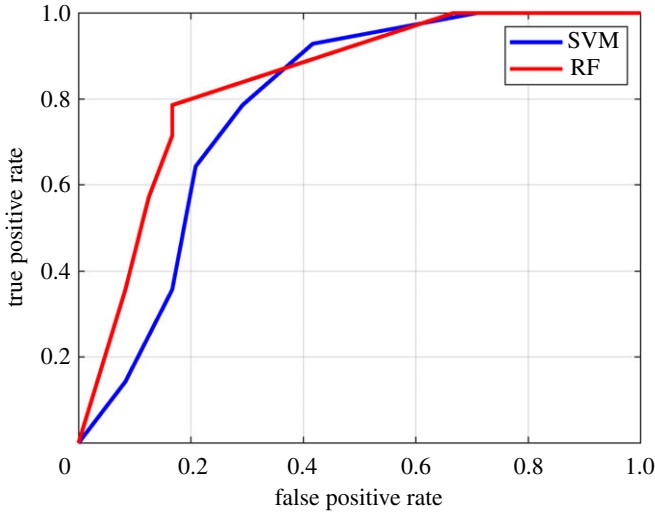

**Figure 7.** Receiver operating characteristics (ROC) curves for RF and SVM classifiers.

## 4. Discussion

It has been shown that actigraphy is able to identify sleep versus wake in insomnia at acceptable levels [43–45]. However, that is not sufficient to serve as a screening tool. Rather, we need an algorithm that can identify those experiencing insomnia from those with normal sleep. Moreover, prior studies validated their algorithm using single night actigraphy signals (in a laboratory environment) and did not focus on multi-night recordings (in a home environment) for assessing insomnia [43,46,47]. Our study appears to be the first in the automated detection of CI from the multi-night actigraphic signal. In this regard, we believe the overall accuracy of 80% with 76% sensitivity and 82% specificity is an excellent start towards a data-driven automated actigraphic analysis framework for the detection of individuals with CI using data collected in an ambulatory setting.

Furthermore, the fact our healthy population were the bed partners of the subjects with insomnia is a strength of our design. It has been shown one bed partner affects the sleep of the other, especially with respect to passing wake episodes between them [15,24]. Given that, the performance (accuracy, sensitivity and specificity) reported here are very encouraging as the first steps. Ultimately, it will be important for any pre-screening tool based on objective sleep to be able to identify insomnia independent of any influence from a bed partner. Outcomes of the study indicate that there is potential for developing an actigraphic-based insomnia pre-screening device that can be deployed in research and clinical settings. Moreover, given the proliferation of consumer sleep trackers using motion data to assess sleep, there is the potential for such a pre-screening tool, once further developed and validated, to be deployed on a community level. This would, of course, require manufacturers to either provide open access to the motion data (which some currently do) or incorporate such an algorithm into their proprietary system.

### 4.1. Feature values and patterns

We have found that values of amplitude (mean) and variability parameters (s.d., SD1, SD2) are consistently high across all intensity filter levels in individuals with CI compared to healthy sleepers. This aligns with our previous findings in subjects with acute insomnia [17], where we found that increasing amounts of activity, due to longer or more frequent waking periods in the nocturnal actigraphy data, contributed to the higher amplitude of statistical measures. Among nonlinear parameters, ratio and SampEn showed an opposite trend to each other across all intensity filtering levels. SampEn values are higher for individuals with CI, suggesting that the irregularity in nocturnal actigraphy traces in those with CI is higher than healthy sleepers. Interestingly, the differences in irregularity values increase with increasing intensity filtering levels, which can be attributed to the removal of noisy or subtle motor activities. Among the sleep parameters, WASO, SWR and SL showed higher values for individuals with CI compared to healthy sleepers except for $InF_1 = 0$. This shows that low motor activity often leads to misclassification of wake recognition, contributing to inaccurate WASO, SWR and SL values [48]. On the contrary, SE and TST are constantly higher for healthy individuals in comparison to those with CI. These findings align

with other reported sleep laboratory studies [15,16]. Differentiating sleep and wake with the absence of movement using a single wrist-worn actigraphy sensor is a long-standing problem. To do this may require multimodal systems that can monitor vital physiological signals in addition to motor activity [49]. However, in this study, we have restricted our approach to a single wrist-worn sensor to allow a less obtrusive long-term monitoring alternative for pre-screening individuals for insomnia. The maximum AUC values (0.72, 0.70, 0.70 and 0.70) were found for features mean, SD1, (SD1, SWR) and SD1 across intensity filter levels from 0 to 80, respectively (table 4). Interestingly, SD1 showed AUC values of 0.70 across all intensity filtering levels, which shows that the distinguishing capability of SD1 is not affected by the intensity filtering. This indicates that Poincaré-derived features are not affected by noisy or subtle motor activity and are good markers of the wake activity patterns in actigraphy signal. This is also corroborated by our previous studies, where we used these features to quantify biomarkers from physiological and movement signals [30,32]. Conversely, statistical measures of activity like mean and s.d. are found to be sensitive to the level of intensity filtering and their AUC decreases with increasing filtering levels (indicated by the blue asterisk in figure 4). This trend might be attributed to the fact that measures such as mean and s.d. are influenced by activities such as subtle movements, which get removed after the application of intensity filtering. Similarly, sleep parameters like WASO, SWR and SE show an improved AUC as they are affected by the subtle motor activity registered on the actigraphy signal. Sleep parameters (WASO, SWR and SE) and SampEn-derived features show an opposite trend as the feature AUC increased with increasing filtering levels (indicated by the red asterisk in figure 4). Although SampEn is not affected by observational noise in the signal, the increase in AUC with the application of increased intensity filtering can be attributed to removal of noisy or subtle motor activity, which makes the high-amplitude activity more apparent in individuals with CI (figure 2). It is known that the presence of noise in the time series increases the SampEn value [50]. Thus, the decreasing values of the SampEn feature with increasing filtering levels in both groups (CI and healthy sleepers) also support this finding. The changes in the feature and AUC values with varying intensity filtering highlight the significance of such filtering approaches on actigraphy data. In addition, such variations indicate that certain features are measures of activity, which is positively associated with the wake time. Thus, the feature relevance varies as the level of intensity filtering is changed. These findings will help further the research for deducing efficient markers of nocturnal awakenings from actigraphy signals that will, in turn, lead to an improved evaluation of insomnia subjects using actigraphy.

Furthermore, accurate wake recognition from actigraphy data is challenging due to the operational deficits of actigraphy [48]. In addition, features derived from actigraphy data may be affected by the sensitivity of the recording device to the level of motor activity due to different manufactures and proprietary software. However, the application of intensity filtering at different levels (InF$_i$ = {0, 20, 40, 80}, $i$ = 1, 2, 3, 4) and the features derived from the filtered actigraphy signals (table 4) reduces the bias of motor activities and the effect of inaccurate wake recognition. Thus, the proposed intensity filtering approach that combines features at different intensity filtering levels affords a more robust signature of CI from nocturnal actigraphy data. The feature AUC values suggest that no single feature can be used as a stand-alone marker for distinguishing individuals with CI from healthy sleepers. This warrants the use of nonlinear machine learning techniques and multiple features to build a CI detection model. In our study, we assigned a good night label to all healthy individuals and a bad night label to all individuals with CI. However, in reality, a healthy sleeping individual can have a few bad nights of sleep during the monitoring period while the individual with CI can have a few good nights of sleep. This partially describes the reason behind each group's higher variance of feature values, and lower AUC value (table 4 and figure 3).

## 4.2. Classification of chronic insomnia

Although the distribution/histograms of raw signals for subjects with CI show a wider distribution in comparison to healthy individuals (figure 2), this trend is not consistent for all the monitored nights and some nights for healthy and CI individuals were found to have similar distribution (figure 2d,e). This highlights the significance of multi-night data in developing automated models for differentiating individuals CI from healthy sleepers. Our study proposes a first automated approach using multi-night data from a wrist-worn actigraphy unit for automated detection of individuals with CI. The proposed automated approach has utility for in-home monitoring and pre-screening of CI in a less obtrusive manner than in-laboratory PSG.

Since the current study uses only actigraphy data collected from the out-of-laboratory environment, the night-level labelling was not available. Instead, we proposed a data-driven method for predicting the quality of a night's sleep (good/bad) by considering all nights from healthy individuals as good nights

(label 0) and from individuals with CI as bad nights (label 1). The model showed a night-level classification accuracy of 73% (table 5a); however, a higher subject-level classification accuracy of 80% is achieved using the RF classifier with the optimally deduced night threshold of Th = 4 (table 5b). The lower night-level performance can be attributed to the fact that not all healthy controls will have good nights, and not all individuals with CI will have bad nights [45]. However, the wake signatures over multi-night monitoring are still different for healthy sleepers and individuals with CI as seen from the higher subject-level classification performance after employing the optimal night threshold Th = 4 (table 5). It is interesting to note that the optimal learned night threshold for classifying an individual as having CI is Th = 4 (i.e. at least four models predicted bad nights for an individual to be classified as CI) for both SVM and RF classifier (table 5b). The proposed model showed an overall accuracy of 80% corresponding to the optimal threshold Th = 4 using the RF classifier (figure 5). This shows that our model can efficiently capture robust signatures of CI from multi-night wrist-actigraphy data. It is also worth noting the four-night threshold we identified is very close to the three-night threshold of subjective symptoms contained within the DSM-5 and ICSD-3 diagnostic criteria for insomnia disorder.

In addition, the effect of recording length (total nights) on subject-level classification performance is also assessed by varying the number of analysed nocturnal actigraphy recording length from three to seven nights. The classification performance decreased as the number of total analysed nights decreased (figure 6). Based on the performance of the classifier (figure 6), it would be safe to assume that a minimum of five nights of recording is required to build an automated model for performance better than a random guess. This is further supported by the histogram distribution of the raw nocturnal actigraphy signals for individuals with CI and healthy sleepers (figure 2). Thus the proposed model can be used to extract a robust signature of CI over multi-night monitoring by combining and optimizing actigraphy-derived features.

The performance of the proposed model (accuracy: 80%, sensitivity: 76%, specificity: 82%) is a promising start for a research-grade actigraphy device for automated pre-screening of individuals with insomnia in their home environment. The performance of the model is especially good, considering that healthy sleepers can be affected by bed partners with insomnia [45]. This is also a limitation of the study as bed partners were used as controls. This may further explain the reason behind the relatively lower accuracy (80%) of this model compared to our acute insomnia detection model (84%) [17]. In addition, the specificity (82%) of this model is comparable with pooled specificity (85%) of insomnia severity index (ISI)-based insomnia detection [51]. However, the sensitivity (76%) is lower than the pooled sensitivity (88%), which may be attributed to the limitations of using bed partners as control [51].

It has been previously shown that bed partners affect each other's sleep [15,45]; however, the overall model performance (accuracy: 80%, sensitivity: 76%, specificity: 82%) highlights the efficacy of the proposed approach and the significance of multi-night monitoring for automated decision-making, which is seldom seen in actigraphy research. Further, the proposed model will add to the current state-of-the-art insomnia research by providing an objective data-driven assessment method to enhance the decision support for clinicians and sleep researchers with potential utility as a pre-screening device.

## 4.3. Effect of randomization

Our model assesses the quality of each night sleep over the seven nights of monitoring to make an informed decision. To highlight the significance of multi-night monitoring and to further validate the hypothesis used for night labelling, we have generated a model using random night labels. Notably, the performance of the model with random night labels is close to a random guess (accuracy—RF: 56%, SVM: 48%), in contrast with the accuracy of 80% (sensitivity: 76%, specificity: 82%). This signifies the validity of this approach, in that healthy sleepers will have a relatively higher number of good night sleeps compared to those with CI. In addition, the findings of the study suggest that the use of advanced machine learning techniques with multi-night actigraphy data can capture the signature of CI even in the presence of noise (inherent error) in the labelling.

## 4.4. Conclusions and future work

In conclusion, the classification performance of the proposed model (accuracy: 80%, sensitivity: 76%, specificity: 82%) shows the potential for actigraphy device-based pre-screening of individuals for CI. The RF classifier showed a better performance in comparison to SVM and thus would be more suitable for further development (figure 7). The findings from the study (figure 6) suggest that an accuracy greater

than 60% could be achieved by using three nights of data, which is not sufficient for accurate classification. Furthermore, reducing the number of nights to less than five nights increases the chances for misclassification since insomnia individuals can sleep well in random nights. These findings align with the guidelines for insomnia diagnosis and treatment by the European Sleep Research Society [52]. In addition, the ability to establish markers of CI from multi-night actigraphy signals has the potential to promote further in-home monitoring devices and applications for mass screening. A home-based monitoring device will provide a patient-centric solution by allowing individuals to take greater control of their health. Moreover, a pre-screening device will help increase awareness about sleep disorders as most individuals with insomnia do not seek medical attention for their condition.

Furthermore, it is possible that certain healthy individuals might also have undiagnosed insomnia. Therefore, the performance of the proposed approach might further improve if an association with clinically determined insomnia indices can be deduced from actigraphy data. As a future study, we would investigate actigraphy-derived markers that can correlate with clinically determined insomnia indices. Incorporation of such indices within the current model will allow for quantitative feedback to the patient, clinicians and sleep researchers in real time.

Ethics. The study was approved by Monash University Human Research Ethics Committee, approval no. CF16/276 - 2016000125. All participants signed informed consent.

Data accessibility. The dataset used in this paper and the codes, which produce all of the results, included in the paper, are available from the Dryad Digital Repository at: https://doi.org/10.5061/dryad.b8gtht7bh [53].

Authors' contributions. S.K. contributed to the data preparation, data analysis, software and writing of the paper. C.K. contributed to feature engineering, data analysis, software and writing of the paper. Y.Z. contributed to machine learning models, software and writing of the paper. S.S. contributed to data analysis and writing of the paper. S.P.A.D. contributed to data collection and writing of the paper. J.G.E. contributed to the writing of the paper. M.A. contributed to the design of the study, methods for analysis and writing of the paper.

Competing interests. We declare we have no competing interests.

Funding. Study was supported by National Health and Medical Research Council (NHMRC) grant # APP1105458 (SPAD).

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
