## [Peer Review File · Royal Society Open Science]

Review History

RSOS-202264.R0 (Original submission)

Review form: Reviewer 1 (Sungkyu Shaun Park)

Is the manuscript scientifically sound in its present form?

No

Are the interpretations and conclusions justified by the results?

No

Is the language acceptable?

Yes

Do you have any ethical concerns with this paper?

No

Have you any concerns about statistical analyses in this paper?

No

Recommendation?

Major revision is needed (please make suggestions in comments)

Comments to the Author(s)

The current paper provides an intriguing two-phase model to predict the quality of a night's sleep and to accordingly detect patients suffering from chronic insomnia (CI). By collecting 7 consecutive data from 40 CI patients and 40 healthy partners and by utilizing Random Forest, they could get 80% of accuracy on classifying CI subjects.

Major Comments:

- I like the concept of the 2nd phase of the proposed model. However, I have some concerns about the 1st phase. Most importantly, I believe that all participants have their own baselines with the extracted features, respectively, but the authors ran the leave-one-out cross-validation for the entire dataset regardless of the subjects. In this light, I think normalizing each feature by subject is inevitable before putting it in the model. If possible, I recommend consulting this issue with a domain expert like a psychiatrist.

- For the 1st phase, I feel uncomfortable that labeling the quality of sleep as bad for all nights of the CI patients while labeling it as good for all nights of the healthy partners. Can we label like this way to CI patients since they are on a chronic status? Then, how can we differentiate a chronic patient and an acute patient? If possible, I recommend consulting this issue with a domain expert like a psychiatrist, too. Since the ground-truth labels can be wrong, I am not entirely sure how much I can trust the prediction results.

- It would be good if the authors could include more state-of-the-art or currently off-the-shelf machine-/deep-learning algorithms on their predictive model. It seems Random Forest and SVM are out-dated ones to use and to compare for now. For your information, please refer to the references below as examples:

[1] Haghayegh, S., Khoshnevis, S., Smolensky, M. H., & Diller, K. R. (2020). Application of deep learning to improve sleep scoring of wrist actigraphy. *Sleep Medicine*, 74, 235-241.

[2] Park, S., Li, C. T., Han, S., Hsu, C., Lee, S. W., & Cha, M. (2019, July). Learning sleep quality from daily logs. In *Proceedings of the 25th ACM SIGKDD International Conference on Knowledge Discovery & Data Mining* (pp. 2421-2429).

[3] Chambon, S., Galtier, M. N., Arnal, P. J., Wainrib, G., & Gramfort, A. (2018). A deep learning architecture for temporal sleep stage classification using multivariate and multimodal time series. *IEEE Transactions on Neural Systems and Rehabilitation Engineering*, 26(4), 758-769.

- Also, I'd like to know more about the validity of collecting healthy partners' data together. Intuitively, it seems like a reasonable approach but at the same time, I am concerned about the possible complex entanglement between the two subjects, e.g., how a healthy partner's sleep is different from a healthy individual who does not sleep with a CI patient? I'd like to hear more of the underline notions on the research design.

Minor Comments:

In addition, there are some minor comments below.

- The authors mentioned that they make patients wear the devices 24/7. In this regard, I am wondering why they did not use the collected day-time features.

- Some figures are hard to read. Labels are too small to read, and there are no clear definitions on the x-axis and y-axis.

- I'd like to know more about the used device's information. e.g, model name.

- Typo: In lines 41-42, I think "(iii) Sensitivity" should be "(iii) Accuracy." There are many unnecessary blanks to be erased, e.g., in line 8, a blank before "TIB" should be erased.

Review form: Reviewer 2 (Mario Miguel)

Is the manuscript scientifically sound in its present form?

No

Are the interpretations and conclusions justified by the results?

Yes

Is the language acceptable?

Yes

Do you have any ethical concerns with this paper?

Yes

Have you any concerns about statistical analyses in this paper?

Yes

Recommendation?

Accept with minor revision (please list in comments)

Comments to the Author(s)

The manuscript entitled "A machine learning model for multi-night actigraphic detection of chronic insomnia: development and validation of a pre-screening tool", from Kusmakar and colleagues, proposed a machine-learning approach to objectively classify chronic insomnia patients from healthy partners, based on bedtime actigraphy data. First, I would like to thank the Authors for letting the code and the data available for readers.

In my view, one of the clearest strengths of the approach used in this manuscript is the idea of performing the actigraphy feature extraction not based on 30 seconds or 1-minute epochs (sleep-wake determination) to resemble the sleep-wake PSG gold-standard epochs, which is usually an unfruitful and outdated effort to prove that actigraphy can be used to infer sleep/wake perfectly. Instead, the Authors developed a new signal processing pipeline to extract time-domain dynamic and non-linear features from actigraphy data in a multi-night design. Although partially dependent on the proprietary algorithm (from Respiroics) to obtain features such as TIB, TST, SE, SL, WASO, the inclusion of other time-series variables, such as sample entropy and Pincare' plots features, makes it easier to reproduce the results and to realise how the results obtained here can be confronted with data/results from different settings by other research groups. Another clear strength here is how data was objectively obtained without input from the participants or the usage of sleep diaries, which provides the ability to achieve large-scale data collection. Lastly, using CI patients and their partners was a clever design. However, apart from the authors' clear strengths in this manuscript, some questions need to be further clarified to increase the capacity of understanding and interpretability of the findings presented here.

1. In the introduction section, although absolutely well organised, I would suggest the Authors include a reflection on why ordinary extracted sleep features (TST, SE, SL, WASO) are not enough to feed a machine-learning algorithm to distinguish insomnia patients from healthy partners, especially considering that clinicians are familiarised with these variables and not with entropy, for instance, which would make the results less intuitive.
2. In the methodology section, there is no further description of the selection of the 7 day-time series. I mean that a 7-day long series includes weekdays and weekends, and this composition could impact the layer-1 machine learning model, increasing the "noise" level. Moreover, in the SVM model description, I could not find the details of the estimation of the parameter C, only the indication that a linear kernel was adopted. Have you performed a grid search? Regarding the random forest model, how the exact number of trees was determined? In

terms of parameter tuning, have you used the random grid to search for the best hyperparameters?

3. It seems to be a problem in the description of the performance matrix. Instead of accuracy (iii), it is written sensitivity.

4. It was difficult for me to understand part of the study design, particularly layer one of the machine learning approaches. Based on the text and table IV, it seems that the Authors calculated the AUCs of single feature classifiers, instead of a model comprising the entire list of potential predictors, which means that 12 (number of features) X 4 (thresholds, InF 0,20,40,80) distinct models were tested in layer one, correct? If so, what is exactly displayed in table Va, the best-fit model?

5. Discussion section: when the Authors described that low motor activity often leads to misclassification of wake recognition, contributing to inaccurate WASO, SWR, and SL values, they are absolutely right. I would suggest them to include a very succinct description of how difficult it is to discriminate between sleep and wake + absence of movement, a problem that has been following actigraphy specialists since the 1970s. One possible way to overcome this limitation of actigraphy is the adoption of additional signals, such as light exposition and temperature. Taking advantage of this theme, I would also suggest the discussion of the results from Beatriz Rodriguez-Morilla et al. (10.3389/fnins.2019.01318), whose results demonstrate that the inclusion of temperature and light improves the performance of a single tree model in discriminating controls, insomnia and DSPD patients. Moreover, on page 13, line 20, the authors state that they employed classifiers that incorporate the inter-relationship between multiple independent variables. However, it was also not possible for me to find in the text where the description of the inter-relationship between the multiple variables was described. Which metrics were adopted? Which tests were performed?

Review form: Reviewer 3

Is the manuscript scientifically sound in its present form?

Yes

Are the interpretations and conclusions justified by the results?

Yes

Is the language acceptable?

Yes

Do you have any ethical concerns with this paper?

No

Have you any concerns about statistical analyses in this paper?

No

Recommendation?

Accept with minor revision (please list in comments)

Comments to the Author(s)

The present manuscript studies actigraphy data of 7 successive nights in subjects with insomnia and their bed partners and applies machine learning techniques in order to establish a model that automatically distinguishes between normal sleepers and subjects with insomnia. The model was successfully applied in a previous publication for acute insomnia in a smaller population (45

subjects) and is applied in the present manuscript in a larger population with chronic insomnia (80 subjects).

The manuscript is well written, with a clear methodology and results and also the discussion and conclusions make sense. I recommend the manuscript for publication after the authors respond to a few minor questions on general and specific aspects of the manuscript.

General question:

- The accuracy of the original model for acute insomnia in 45 subjects was 84% with a sensitivity of 76% and a specificity of 92%, whereas the present model for chronic insomnia in 80 subjects is 80% with a sensitivity of 76% and a specificity of 82%. Can the authors say something on the comparison between the numbers for accuracy, sensitivity and specificity for both models? How do these numbers compare to alternative classification methods such as questionnaires, diaries, PSG, etc.?

Specific questions:

[Abstract-Methods]: "bilateral nocturnal actigraphy signals"

- What is meant with "bilateral"? Do subjects wear 1 or 2 watches? In the case of 1 wrist, which wrist was chosen, the dominant wrist?
 - "Nocturnal actigraphy signals". Why has the full 24h of data per day not been considered here, would the daypart not give additional information on insomnia?

[II. Methods, E. Feature extraction]:

- Notation "SD" vs. "sd" in the rest of the manuscript
 - Area and CCM parameters are not explicitly explained in the text
 - eq. (2) uses the extra parameter of "time in bed (TIB)" but the manuscript does not explain how it was calculated
 - One could think of additional nonlinear and sleep-specific features than the ones reported in the text, how would including more/less features affect the predictive capacities of the model?
 - 12 different features evaluated at 4 different lnF levels would add up to a feature set of $48 = 12 * 4$ exclusive features, and not $40 = 10 * 4$ as stated in the manuscript? I have a similar doubt on the feature set of the previous publication on acute insomnia which reports a feature set of $40 = 10 * 4$, whereas I count $44 = 4 * 11$ features?

[III. Results, B. Classification of Chronic insomnia]

- The manuscript says that "the model performance decreases with the length (total analysed nights) (Fig. 6)". I guess this is a typo, and the text should say "increases", perhaps an additional comment can be added to the caption of Fig. 6 noting that the direction of the horizontal axis is opposite to the standard direction.
 - "the optimal classification performance is achieved for 7 nights of recordings and the model classification performance drops to a random guess for a recording below 5 nights". If you would have longer recordings, e.g., 2 weeks, would it be possible to improve still the classification? Doesn't Fig. 6 suggest that accuracy stays >60% even when using recordings of only 3 nights, and therefore well above the 50% of a random guess?

[IV. Discussion]

- It is known that noise in a time series increases SampEn. Can you give a reference?

Decision letter (RSOS-202264.R0)

Dear Dr Angelova

The Editors assigned to your paper RSOS-202264 "A machine learning model for multi-night actigraphic detection of chronic insomnia: development and validation of a pre-screening tool" have now received comments from reviewers and would like you to revise the paper in accordance with the reviewer comments and any comments from the Editors. Please note this decision does not guarantee eventual acceptance.

Please submit your revised manuscript and required files (see below) no later than 21 days from today's (ie 13-Apr-2021) date. Note: the ScholarOne system will 'lock' if submission of the revision is attempted 21 or more days after the deadline. If you do not think you will be able to meet this deadline please contact the editorial office immediately.

on behalf of Dr Mirco Musolesi (Associate Editor) and Marta Kwiatkowska (Subject Editor)
openscience@royalsociety.org

Associate Editor Comments to Author (Dr Mirco Musolesi):
Comments to the Author:

The reviewers pointed out some issues with the current version of the article that need to be addressed both in terms of methodology and presentation of the results. For this reason, I would recommend a major revision of the article. I would invite the authors to carefully address all the points listed in the reviews.

Reviewer comments to Author:

Reviewer: 1

Comments to the Author(s)

The current paper provides an intriguing two-phase model to predict the quality of a night's sleep and to accordingly detect patients suffering from chronic insomnia (CI). By collecting 7 consecutive data from 40 CI patients and 40 healthy partners and by utilizing Random Forest, they could get 80% of accuracy on classifying CI subjects.

Major Comments:

- I like the concept of the 2nd phase of the proposed model. However, I have some concerns about the 1st phase. Most importantly, I believe that all participants have their own baselines with the extracted features, respectively, but the authors ran the leave-one-out cross-validation for the entire dataset regardless of the subjects. In this light, I think normalizing each feature by subject is inevitable before putting it in the model. If possible, I recommend consulting this issue with a domain expert like a psychiatrist.

- For the 1st phase, I feel uncomfortable that labeling the quality of sleep as bad for all nights of the CI patients while labeling it as good for all nights of the healthy partners. Can we label like this way to CI patients since they are on a chronic status? Then, how can we differentiate a chronic patient and an acute patient? If possible, I recommend consulting this issue with a domain expert like a psychiatrist, too. Since the ground-truth labels can be wrong, I am not entirely sure how much I can trust the prediction results.

- It would be good if the authors could include more state-of-the-art or currently off-the-shelf machine-/deep-learning algorithms on their predictive model. It seems Random Forest and SVM are out-dated ones to use and to compare for now. For your information, please refer to the references below as examples:

[1] Haghayegh, S., Khoshnevis, S., Smolensky, M. H., & Diller, K. R. (2020). Application of deep learning to improve sleep scoring of wrist actigraphy. *Sleep Medicine*, 74, 235-241.

[2] Park, S., Li, C. T., Han, S., Hsu, C., Lee, S. W., & Cha, M. (2019, July). Learning sleep quality from daily logs. In *Proceedings of the 25th ACM SIGKDD International Conference on Knowledge Discovery & Data Mining* (pp. 2421-2429).

[3] Chambon, S., Galtier, M. N., Arnal, P. J., Wainrib, G., & Gramfort, A. (2018). A deep learning architecture for temporal sleep stage classification using multivariate and multimodal time series. *IEEE Transactions on Neural Systems and Rehabilitation Engineering*, 26(4), 758-769.

- Also, I'd like to know more about the validity of collecting healthy partners' data together. Intuitively, it seems like a reasonable approach but at the same time, I am concerned about the possible complex entanglement between the two subjects, e.g., how a healthy partner's sleep is different from a healthy individual who does not sleep with a CI patient? I'd like to hear more of the underline notions on the research design.

Minor Comments:

In addition, there are some minor comments below.

- The authors mentioned that they make patients wear the devices 24/7. In this regard, I am wondering why they did not use the collected day-time features.

- Some figures are hard to read. Labels are too small to read, and there are no clear definitions on the x-axis and y-axis.

- I'd like to know more about the used device's information. e.g, model name.

- Typo: In lines 41-42, I think "(iii) Sensitivity" should be "(iii) Accuracy." There are many unnecessary blanks to be erased, e.g., in line 8, a blank before "TIB" should be erased.

Reviewer: 2

Comments to the Author(s)

The manuscript entitled "A machine learning model for multi-night actigraphic detection of chronic insomnia: development and validation of a pre-screening tool", from Kusmakar and colleagues, proposed a machine-learning approach to objectively classify chronic insomnia patients from healthy partners, based on bedtime actigraphy data. First, I would like to thank the Authors for letting the code and the data available for readers.

In my view, one of the clearest strengths of the approach used in this manuscript is the idea of performing the actigraphy feature extraction not based on 30 seconds or 1-minute epochs (sleep-wake determination) to resemble the sleep-wake PSG gold-standard epochs, which is usually an unfruitful and outdated effort to prove that actigraphy can be used to infer sleep/wake perfectly. Instead, the Authors developed a new signal processing pipeline to extract time-domain dynamic and non-linear features from actigraphy data in a multi-night design. Although partially dependent on the proprietary algorithm (from Respiroics) to obtain features such as TIB, TST, SE, SL, WASO, the inclusion of other time-series variables, such as sample entropy and Pincare' plots features, makes it easier to reproduce the results and to realise how the results obtained here can be confronted with data/results from different settings by other research groups. Another clear strength here is how data was objectively obtained without input from the participants or the usage of sleep diaries, which provides the ability to achieve large-scale data collection. Lastly, using CI patients and their partners was a clever design. However, apart from the authors' clear strengths in this manuscript, some questions need to be further clarified to increase the capacity of understanding and interpretability of the findings presented here.

1. In the introduction section, although absolutely well organised, I would suggest the Authors include a reflection on why ordinary extracted sleep features (TST, SE, SL, WASO) are not enough to feed a machine-learning algorithm to distinguish insomnia patients from healthy partners, especially considering that clinicians are familiarised with these variables and not with entropy, for instance, which would make the results less intuitive.
2. In the methodology section, there is no further description of the selection of the 7 day-time series. I mean that a 7-day long series includes weekdays and weekends, and this composition could impact the layer-1 machine learning model, increasing the "noise" level. Moreover, in the SVM model description, I could not find the details of the estimation of the parameter C, only the indication that a linear kernel was adopted. Have you performed a grid search? Regarding the random forest model, how the exact number of trees was determined? In terms of parameter tuning, have you used the random grid to search for the best hyperparameters?
3. It seems to be a problem in the description of the performance matrix. Instead of accuracy (iii), it is written sensitivity.
4. It was difficult for me to understand part of the study design, particularly layer one of the machine learning approaches. Based on the text and table IV, it seems that the Authors calculated the AUCs of single feature classifiers, instead of a model comprising the entire list of potential predictors, which means that 12 (number of features) X 4 (thresholds, InF 0,20,40,80) distinct models were tested in layer one, correct? If so, what is exactly displayed in table Va, the best-fit model?
5. Discussion section: when the Authors described that low motor activity often leads to misclassification of wake recognition, contributing to inaccurate WASO, SWR, and SL values, they are absolutely right. I would suggest them to include a very succinct description of how difficult it is to discriminate between sleep and wake + absence of movement, a problem that has been following actigraphy specialists since the 1970s. One possible way to overcome this limitation of actigraphy is the adoption of additional signals, such as light exposition and temperature. Taking advantage of this theme, I would also suggest the discussion of the results from Beatriz Rodriguez-Morilla et al. (10.3389/fnins.2019.01318), whose results demonstrate that the inclusion of temperature and light improves the performance of a single tree model in

discriminating controls, insomnia and DSPD patients. Moreover, on page 13, line 20, the authors state that they employed classifiers that incorporate the inter-relationship between multiple independent variables. However, it was also not possible for me to find in the text where the description of the inter-relationship between the multiple variables was described. Which metrics were adopted? Which tests were performed?

Reviewer: 3

Comments to the Author(s)

The present manuscript studies actigraphy data of 7 successive nights in subjects with insomnia and their bed partners and applies machine learning techniques in order to establish a model that automatically distinguishes between normal sleepers and subjects with insomnia. The model was successfully applied in a previous publication for acute insomnia in a smaller population (45 subjects) and is applied in the present manuscript in a larger population with chronic insomnia (80 subjects).

The manuscript is well written, with a clear methodology and results and also the discussion and conclusions make sense. I recommend the manuscript for publication after the authors respond to a few minor questions on general and specific aspects of the manuscript.

General question:

- The accuracy of the original model for acute insomnia in 45 subjects was 84% with a sensitivity of 76% and a specificity of 92%, whereas the present model for chronic insomnia in 80 subjects is 80% with a sensitivity of 76% and a specificity of 82%. Can the authors say something on the comparison between the numbers for accuracy, sensitivity and specificity for both models? How do these numbers compare to alternative classification methods such as questionnaires, diaries, PSG, etc.?

Specific questions:

[Abstract-Methods]: “bilateral nocturnal actigraphy signals”

- What is meant with “bilateral”? Do subjects wear 1 or 2 watches? In the case of 1 wrist, which wrist was chosen, the dominant wrist?
 - “Nocturnal actigraphy signals”. Why has the full 24h of data per day not been considered here, would the daypart not give additional information on insomnia?

[II. Methods, E. Feature extraction]:

- Notation “SD” vs. “sd” in the rest of the manuscript
 - Area and CCM parameters are not explicitly explained in the text
 - eq. (2) uses the extra parameter of “time in bed (TIB)” but the manuscript does not explain how it was calculated
 - One could think of additional nonlinear and sleep-specific features than the ones reported in the text, how would including more/less features affect the predictive capacities of the model?
 - 12 different features evaluated at 4 different InF levels would add up to a feature set of $48 = 12 * 4$ exclusive features, and not $40 = 10 * 4$ as stated in the manuscript? I have a similar doubt on the feature set of the previous publication on acute insomnia which reports a feature set of $40 = 10 * 4$, whereas I count $44 = 4 * 11$ features?

[III. Results, B. Classification of Chronic insomnia]

- The manuscript says that “the model performance decreases with the length (total analysed nights) (Fig. 6)”. I guess this is a typo, and the text should say “increases”, perhaps an additional

comment can be added to the caption of Fig. 6 noting that the direction of the horizontal axis is opposite to the standard direction.

- “the optimal classification performance is achieved for 7 nights of recordings and the model classification performance drops to a random guess for a recording below 5 nights”. If you would have longer recordings, e.g., 2 weeks, would it be possible to improve still the classification?

Doesn't Fig. 6 suggest that accuracy stays >60% even when using recordings of only 3 nights, and therefore well above the 50% of a random guess?

[IV. Discussion]

- It is known that noise in a time series increases SampEn. Can you give a reference?

===PREPARING YOUR MANUSCRIPT===

===PREPARING YOUR REVISION IN SCHOLARONE===

Author's Response to Decision Letter for (RSOS-202264.R0)

See Appendix A.

Decision letter (RSOS-202264.R1)

Dear Dr Angelova,

It is a pleasure to accept your manuscript entitled "A machine learning model for multi-night actigraphic detection of chronic insomnia: development and validation of a pre-screening tool" in its current form for publication in Royal Society Open Science. The comments of the reviewer(s) who reviewed your manuscript are included at the foot of this letter.

on behalf of Dr Mirco Musolesi (Associate Editor) and Marta Kwiatkowska (Subject Editor)
openscience@royalsociety.org

Associate Editor Comments to Author (Dr Mirco Musolesi):

Associate Editor

Comments to the Author:

The authors addressed all the points raised by the reviewers in a very convincing way. I found the response letter very clear. The modifications to the original manuscript are exhaustive. I recommend to accept this paper for publication.

Appendix A

Associate Editor Comments to Author (Dr Mirco Musolesi):

Comments to the Author:

The reviewers pointed out some issues with the current version of the article that need to be addressed both in terms of methodology and presentation of the results. For this reason, I would recommend a major revision of the article. I would invite the authors to carefully address all the points listed in the reviews.

Response: We would like to thank the reviewers for their time and useful suggestions. Our responses are below **in blue**. The changes are made to the manuscript in **tracked changes** and the corresponding clean version of the revised manuscript is provided.

Reviewer: 1

Comments to the Author(s)

The current paper provides an intriguing two-phase model to predict the quality of a night's sleep and to accordingly detect patients suffering from chronic insomnia (CI). By collecting 7 consecutive data from 40 CI patients and 40 healthy partners and by utilizing Random Forest, they could get 80% of accuracy on classifying CI subjects.

Major Comments:

1)

I like the concept of the 2nd phase of the proposed model. However, I have some concerns about the 1st phase. Most importantly, I believe that all participants have their own baselines with the extracted features, respectively, but the authors ran .the leave-one-out cross-validation for the entire dataset regardless of the subjects. In this light, I think normalizing each feature by subject is inevitable before putting it in the model. If possible, I recommend consulting this issue with a domain expert like a psychiatrist.

Response: We agree with the reviewer that subjective variation (different baseline conditions of each subject) is an important condition that should be taken care of for improving model development. However, this adds additional complexity during model development as well as in the translational process. For example, it is not certain what are the exact or most important conditions that we should consider. Sometime, age is considered as one of the very important such factor in many cases. Considering all these factors, in machine learning based model development, since a variety of features are fed into the model, we expect the model will be able to generalise based on the diversity in the features. Therefore, at the beginning we try to develop model without considering the baseline variations of the subject. In addition, we believe that taking account of the baseline conditions will have positive impact on the performance rather than a negative one. Moreover, since our study is based on the retrospective data we cannot add additional parameters for normalising the baseline conditions. Therefore, we have no capacity to change the current model development procedure. We would like to note that in our study the two groups were matched on age because of the partner status. Furthermore, the data collection had appropriate exclusion criteria assuring that patient baseline is not significantly different.

Action: We added a clarification about the age-matched groups in Section II, Part C (Protocol), second paragraph.

2)

A) For the 1st phase, I feel uncomfortable that labeling the quality of sleep as bad for all nights of the CI patients while labeling it as good for all nights of the healthy partners. Can we label like this way to CI patients since they are on a chronic status? Then, how can we differentiate a chronic patient and an acute patient? If possible, I recommend consulting this issue with a domain expert like a psychiatrist, too.

Response: We thank the reviewer for raising this concern. The blind labelling of the quality of sleep of healthy and chronic patients as good and bad is motivated by the fact to develop a data driven framework for differentiating healthy sleepers from individuals with CI without using any sleep diaries. Most studies rely on manual labelling of nights (good/bad) using sleep diaries;

however, the use of sleep diaries introduces errors due to incorrect logs, and models developed on such data may have an inherent subjective bias. In addition, as we want to develop a pre-screening tool for insomnia detection, use of sleep diaries for night labelling will impede the purpose. Therefore, in this study we have used the same blinded labelling approach, which was introduced in our previous study (see reference [17], Angelova et al 2020).

We agree that this is not exactly accurate labelling of individual nights as indicated by the reviewer. For example, every night of a CI individual is not a bad sleep night and vice versa. However, the results decisively indicate that the proposed two levels model can rectify that error and provide the expected outcome.

M. Angelova, C. Karmakar, Y. Zhu, S. P. Drummond, and J. Ellis, "Automated method for detecting acute insomnia using multi-night actigraphy data," *IEEE Access*, vol. 8, pp. 74 413–74 422, 2020.

Action: This is discussed in part III. C Randomization.

B) Since the ground-truth labels can be wrong, I am not entirely sure how much I can trust the prediction results.

Response: The night labels are not the ground truth, the diagnosis of participants as CI or healthy sleeper is the ground truth, this diagnosis is done by clinicians. Further, to ensure that the hypothesis of labelling the quality of healthy sleepers' night as good and individual with CI as bad, we recalculated the performance of the proposed model after random labelling of the nights (please refer to Section III. Results, subsection C. Effect of randomization). After random labelling the model had a best accuracy of 56%. In comparison the proposed approach where the night labels are first predicted using our data-driven approach shows a much-improved performance with an accuracy of 80% (see part III. C Randomization). This highlights that the labelling of nights as good/bad captures the inherent patterns in the data.

Action: No action

3) It would be good if the authors could include more state-of-the-art or currently off-the-shelf machine-/deep-learning algorithms on their predictive model. It seems Random Forest and SVM are out-dated ones to use and to compare for now. For your information, please refer to the references below as examples:

[1] Haghayegh, S., Khoshnevis, S., Smolensky, M. H., & Diller, K. R. (2020). Application of deep learning to improve sleep scoring of wrist actigraphy. *Sleep Medicine*, 74, 235-241.

[2] Park, S., Li, C. T., Han, S., Hsu, C., Lee, S. W., & Cha, M. (2019, July). Learning sleep quality from daily logs. In *Proceedings of the 25th ACM SIGKDD International Conference on Knowledge Discovery & Data Mining* (pp. 2421-2429).

[3] Chambon, S., Galtier, M. N., Arnal, P. J., Wainrib, G., & Gramfort, A. (2018). A deep learning architecture for temporal sleep stage classification using multivariate and multimodal time series. *IEEE Transactions on Neural Systems and Rehabilitation Engineering*, 26(4), 758-769.

Response: We acknowledge that deep learning algorithms can fit complicated data distribution and may get more accurate results based on large and various kinds of dataset. However, they are black box methods that lacks interpretation about the variable dependence and decisions.

Furthermore, they need large number of training data to avoid overfitting issues because deep learning models are heavily over-parameterized and can often get to perfect results on training data. In our paper, we only have the actigraphy data from 40 cohabiting couples, which is not enough to train a complex deep learning model. Therefore, we use the traditional Random Forest and SVM to analyse the importance of features extracted from the actigraphy data and identify the important patterns to explain the behaviors of chronic insomnia. Furthermore, the papers [1], [3] use data from several sensors (from PSG or EEG/ECG sensors) in addition to actigraphy, and [2] data collected over 6 continuous weeks (which may not be feasible in normal living), Insomnia Severity Index and logs from Fitbit, while our aim was to use the actigraphy data only as the purpose is to build a pre-screening tool for detection of insomnia.

We want to clarify that although deep learning is a powerful tool, which has shown its suitability in many applications, it does neither invalidate nor replace the existing classical machine learning algorithms, which often perform better on smaller data set and provide better transparent insights on feature space (Dargan et al., 2019, Gaur et al., 2021).

Dargan, S., Kumar, M., Ayyagari, M.R. and Kumar, G., 2019. A survey of deep learning and its applications: A new paradigm to machine learning. Archives of Computational Methods in Engineering, pp.1-22.

Gaur, M., Faldu, K. and Sheth, A., 2021. Semantics of the Black-Box: Can knowledge graphs help make deep learning systems more interpretable and explainable? IEEE Internet Computing, 25(1), pp.51-59.

Action: We have added a justification for using SVM and RF in section and a reference Section II, part F (end of first paragraph).

4) Also, I'd like to know more about the validity of collecting healthy partners' data together. Intuitively, it seems like a reasonable approach but at the same time, I am concerned about the possible complex entanglement between the two subjects, e.g., how a healthy partner's sleep is different from a healthy individual who does not sleep with a CI patient? I'd like to hear more of the underline notions on the research design.

Response: The reviewer raises an important point here. There are advantages and disadvantages to employing bedpartners as the controls in this study. As the reviewer noted, one bedpartner can indeed affect the sleep of the other bedpartner. Our team has published a series of papers examining this very issue in this specific sample (Walters et al, 2020a), and in a sample where neither partner experiences sleep difficulties (Walters et al, 2020b). Overall, actigraphy-measured sleep was extremely similar in the partners of those with insomnia and the individuals in the couples where both partners were normal sleepers. For example, we reported the following values for partners of insomnia and normal sleepers, respectively: Sleep efficiency 79.2% vs 79.0%; Sleep Latency 13.8min vs 13.6 min; Wake After Sleep Onset 71.1min vs 74.7 min. As can be seen, each of those measures is very similar between the two groups. On the other hand, we also reported the partners of insomnia were woken up by their bedpartner slightly more frequently than are the partners of good sleepers. Again, though the differences were subtle (amounting to 0.9 extra awakenings during the night, on average). Of course, those papers did not examine any of the more complex metrics utilised in this study, so it is unknown what influence an insomnia vs good sleeper bedpartner may have on those. In the end, we decided employing the bedpartners are controls, here, provided a robust test of the ability to discriminate between insomnia and good sleepers, precisely because of the potential mutual influences on sleep. If our algorithms can

separate the two groups in this more challenging situation, they are likely to do even better when individuals are truly independent. This point is briefly mentioned in the original draft of the manuscript (see references [15], [46] listed below).

Walters, EM, Phillips, AJK, Boardman, JM, Norton, PJ, Drummond, SPA. Vulnerability and Resistance to Sleep Disruption by a Partner: A Study of Bed-Sharing Couples. *Sleep Health*, **2020**, 6:506-512. 2020. doi: 10.1016/j.sleh.2019.12.005.

Walters, EM, Phillips, AJK, Mellor, A, Hamill, K, Jenkins, MM, Norton, PJ, Baucom, DH, Drummond, SPA. Sleep and Wake are Shared and Transmitted between Individuals with Insomnia and their Bed-Sharing Partners. *SLEEP*. 2020, 43(1):1-12. doi: 10.1093/sleep/zsz206

Action: We have modified section II, Part C (Protocol) to clarify the effect of bed partner as control.

Minor Comments:

In addition, there are some minor comments below.

- The authors mentioned that they make patients wear the devices 24/7. In this regard, I am wondering why they did not use the collected day-time features.

Response: In this study, our aim was to classify only the night features. During the day, different people were involved in different activities as the data was collected in an uncontrolled environment. Since we did not have labels for day-time activities, we cannot remove the subjective bias. Thus, we are only using the night-time data for developing our models. However, in future it would be interesting to explore uses of 24/7 data and see their effect on the model development.

Action: No action

- Some figures are hard to read. Labels are too small to read, and there are no clear definitions on the x-axis and y-axis.

Response/Action: The figures have been improved, labels and axis labels have now been enlarged and the missing labels added.

- I'd like to know more about the used device's information. e.g, model name.

Response: The device used was Respirationics Actiwatch Spectrum Pro, analysed with Actiware software (Respirationics, Bend, OR)

Action: This has been already mentioned in the text (Section II, Part C, page 5) and further clarified in the revised manuscript in section II. Part C, second paragraph.

- Typo: In lines 41-42, I think "(iii) Sensitivity" should be "(iii) Accuracy." There are many unnecessary blanks to be erased, e.g., in line 8, a blank before "TIB" should be erased.

Response: Thanks for pointing this out.

Action: The typos have been corrected.

Reviewer: 2

Comments to the Author(s)

The manuscript entitled “A machine learning model for multi-night actigraphic detection of chronic insomnia: development and validation of a pre-screening tool”, from Kusmakar and colleagues, proposed a machine-learning approach to objectively classify chronic insomnia patients from healthy partners, based on bedtime actigraphy data. First, I would like to thank the Authors for letting the code and the data available for readers.

In my view, one of the clearest strengths of the approach used in this manuscript is the idea of performing the actigraphy feature extraction not based on 30 seconds or 1-minute epochs (sleep-wake determination) to resemble the sleep-wake PSG gold-standard epochs, which is usually an unfruitful and outdated effort to prove that actigraphy can be used to infer sleep/wake perfectly. Instead, the Authors developed a new signal processing pipeline to extract time-domain dynamic and non-linear features from actigraphy data in a multi-night design. Although partially dependent on the proprietary algorithm (from Respironics) to obtain features such as TIB, TST, SE, SL, WASO, the inclusion of other time-series variables, such as sample entropy and Pincare’plots features, makes it easier to reproduce the results and to realise how the results obtained here can be confronted with data/results from different settings by other research groups. Another clear strength here is how data was objectively obtained without input from the participants or the usage of sleep diaries, which provides the ability to achieve large-scale data collection. Lastly, using CI patients and their partners was a clever design. However, apart from the authors' clear strengths in this manuscript, some questions need to be further clarified to increase the capacity of understanding and interpretability of the findings presented here.

1. In the introduction section, although absolutely well organised, I would suggest the Authors include a reflection on why ordinary extracted sleep features (TST, SE, SL, WASO) are not enough to feed a machine-learning algorithm to distinguish insomnia patients from healthy partners, especially considering that clinicians are familiarised with these variables and not with entropy, for instance, which would make the results less intuitive.

Response: Machine learning algorithms like RF and SVM have the capacity to rank the features and use features efficiently based on their discrimination capabilities. Therefore, the model usually provides better performance using larger number of informative features and manual selection of small set of features are unnecessary. Thus, in this study we have not limited to develop model using only the sleep parameters and added other statistical and non-linear features that can be extracted from the actigraphy time-series signal.

Action: We have modified the introduction section (Section I, Paragraph 5).

2.

A)

In the methodology section, there is no further description of the selection of the 7 day-time series. I mean that a 7-day long series includes weekdays and weekends, and this composition could impact the layer-1 machine learning model, increasing the “noise” level.

Response:

- The 7 days were not fixed, the participants could start and stop on any day of the week, in practice they all started on a weekday, and had no restrictions as to which weekday. Each time series included one weekend and 5 working days. Therefore, each participant had similar distribution of “noise level”.

Action: We have added a clarification in Section II, Part C (Protocol).

B)

Moreover, in the SVM model description, I could not find the details of the estimation of the parameter C, only the indication that a linear kernel was adopted. Have you performed a grid search? Regarding the random forest model, how the exact number of trees was determined? In terms of parameter tuning, have you used the random grid to search for the best hyperparameters?

Response: We have used the Sequential Minimal Optimisation (Fan et al., 2005) for best hyperparameters selection of SVM. For the random forest model, we have used the default parameter settings in Matlab i.e., number of bagging trees=100, MaxNumSplits=data size-1, and NumVariablesToSample is the square root of the number of predictors is for random forest. We have applied a 10-fold cross validation method to find the optimal model, which was then used for generating the test results.

Fan, R.E., Chen, P.H., Lin, C.J. and Joachims, T., 2005. Working set selection using second order information for training support vector machines. *Journal of machine learning research*, 6(12).

Action: We have added this information in Section II. Methods, Subsection F.

3. It seems to be a problem in the description of the performance matrix. Instead of accuracy (iii), it is written sensitivity.

Response: We thank the reviewer for pointing this out.

Action: This has been corrected in the revised manuscript, Section II, part G.

4. It was difficult for me to understand part of the study design, particularly layer one of the machine learning approaches. Based on the text and table IV, it seems that the Authors calculated the AUCs of single feature classifiers, instead of a model comprising the entire list of potential predictors, which means that 12 (number of features) X 4 (thresholds, InF 0,20,40,80) distinct models were tested in layer one, correct? If so, what is exactly displayed in table Va, the best-fit model?

Response: Indeed, we calculated the AUCs of individual features to show that a single feature cannot be used as a classifier (Table 4). Then we progressed with a multidimensional model comprising the full list of features, namely 48 (12 x 4) features were used in two classification models, SVM and RF. Table Va shows the performance of these two models at Layer 1 of the machine learning model with the optimised threshold (Th=4).

Action: We have modified the caption of Table V to reflect this and added a sentence at the end of Section III, Part A.

5.

A)

Discussion section: when the Authors described that low motor activity often leads to misclassification of wake recognition, contributing to inaccurate WASO, SWR, and SL values, they are absolutely right. I would suggest them to include a very succinct description of how difficult it is to discriminate between sleep and wake + absence of movement, a problem that has been following actigraphy specialists since the 1970s. One possible way to overcome this limitation of actigraphy is the adoption of additional signals, such as light exposition and temperature. Taking advantage of this theme, I would also suggest the discussion of the results from Beatriz Rodriguez-Morilla et al. (10.3389/fnins.2019.01318), whose results demonstrate that the inclusion of temperature and light improves the performance of a single tree model in discriminating controls, insomnia and DSPD patients.

Response: We thank the reviewer for the interesting comment and have extended this part and included the reference in the revised manuscript.

Action: We added comment in section IV. Discussions, Subsection A. Feature Values and patterns of the revised manuscript, end of first paragraph. We also included the reference (reference [49]) in the list of References.

B)

Moreover, on page 13, line 20, the authors state that they employed classifiers that incorporate the inter-relationship between multiple independent variables. However, it was also not possible for me to find in the text where the description of the inter-relationship between the multiple variables was described. Which metrics were adopted? Which tests were performed?

Response: We thank the reviewer for this point. We did not employ any metric to calculate the inter-relationship of the features. However, the two machine learning algorithms used in the paper inherently utilizes the inter-relationships between individual features to develop an optimal classification model (see reference 36. L. Breiman, "Random forests" in the revised manuscript). Since this sentence raised confusion, we removed it in the revised manuscript.

Action: Removed last sentence Section IV, part A.

Reviewer: 3

Comments to the Author(s)

The present manuscript studies actigraphy data of 7 successive nights in subjects with insomnia and their bed partners and applies machine learning techniques in order to establish a model that automatically distinguishes between normal sleepers and subjects with insomnia. The model was successfully applied in a previous publication for acute insomnia in a smaller population (45 subjects) and is applied in the present manuscript in a larger population with chronic insomnia (80 subjects).

The manuscript is well written, with a clear methodology and results and also the discussion and conclusions make sense. I recommend the manuscript for publication after the authors respond to a few minor questions on general and specific aspects of the manuscript.

General question:

- The accuracy of the original model for acute insomnia in 45 subjects was 84% with a sensitivity of 76% and a specificity of 92%, whereas the present model for chronic insomnia in 80 subjects is 80% with a sensitivity of 76% and a specificity of 82%. Can the authors say something on the comparison between the numbers for accuracy, sensitivity and specificity for both models? How do these numbers compare to alternative classification methods such as questionnaires, diaries, PSG, etc.?

Response: We have discussed a possible reason for the variation of accuracy, sensitivity and specificity due to bed partners are not true healthy sleep controls in the last paragraph of section B in Results. We also compared with available literature based on Insomnia Severity Index (ISI) (see reference [51] Chiu et al 2016). We would like to point out that PSG is not used routinely by clinicians for assessment of insomnia. When PSG is used, it is used only to rule out the presence of other sleep disorders and not to rule in the presence of insomnia (Riemann et al 2017, Cunnington et al 2013).

H. Y. Chiu, L. Y. Chang, Y. J. Hsieh, and P. S. Tsai. (2016). A meta-analysis of diagnostic accuracy of three screening tools for insomnia," *Journal of Psychosomatic Research*, 87, 85-92.

Riemann D. et al. (2017). European guideline for the diagnosis and treatment of insomnia. *J Sleep Res.* 26, 675-700. doi: 10.1111/jsr.12594.

Cunnington, D., Junge, M.F., Fernando, A.T. (2013) Insomnia: prevalence, consequences and effective treatment. *MJA* 199, S8, S36-340. doi: 10.5694/mja13.10718

Action: We expanded the discussion in Section IV, part B, paragraph four.

Specific questions:

[Abstract-Methods]: “bilateral nocturnal actigraphy signals”

- What is meant with “bilateral”? Do subjects wear 1 or 2 watches? In the case of 1 wrist, which wrist was chosen, the dominant wrist?

Response: We thank the reviewer for raising this point. We have removed the word bilateral as one watch was worn on the non-dominant wrist, which is a standard practice in the actigraphy research. The changes are highlighted in the abstract of the revised manuscript.

Action: removed bilateral from the abstract

- “Nocturnal actigraphy signals”. Why has the full 24h of data per day not been considered here, would the daypart not give additional information on insomnia?

Response: We thank the reviewer of raising this point. We agree with the reviewer that it would be interesting to include the day time signals as well, however, as we used retrospective data, daily activities were not labelled. Although there are studies using 24h unannotated data, this is out of the scope of our current study.

Action: no action.

[II. Methods, E. Feature extraction]:

- Notation “SD” vs. “sd” in the rest of the manuscript

Response: We thank the reviewer for pointing this out and it has now been corrected in the revised manuscript.

Action: This has been corrected throughout the manuscript.

- Area and CCM parameters are not explicitly explained in the text

Response: The area and CCM are Poincare derived parameters and considering the scope of the study we have not incorporated detailed explanation; however, we have provided all relevant references for further studying the parameters.

Action: no action.

- eq. (2) uses the extra parameter of “time in bed (TIB)” but the manuscript does not explain how it was calculated

Response: Thanks for raising this point, the Time in Bed (TIB) parameter is used as the actiwatch generated rest interval.

Action: Clarification for TIB has been made in the last paragraph of Section II, part E.

- One could think of additional nonlinear and sleep-specific features than the ones reported in the text, how would including more/less features affect the predictive capacities of the model?

Response: We thank the reviewer for raising this interesting point. In this study we have used only a small subset of time-domain features based on the findings of our previous study (reference [17] of the modified manuscript) and time-domain features have been shown to be the best features for activity recognition (reference [28] of the revised manuscript). Dropping some features might reduce the performance of the model however, adding more non-linear and frequency

domain features can improve the performance only at the cost of increasing the model complexity. In contrast, time-domain features offer low computational complexity which is important in the context of real-time analysis.

Action: no action.

- 12 different features evaluated at 4 different InF levels would add up to a feature set of $48 = 12 * 4$ exclusive features, and not $40 = 10 * 4$ as stated in the manuscript? I have a similar doubt on the feature set of the previous publication on acute insomnia which reports a feature set of $40 = 10 * 4$, whereas I count $44 = 4 * 11$ features?

Response: We thank the reviewer for spotting the typo in the text of the paper. We have extracted $12 * 4$ time-domain features from the nocturnal actigraphy signals, of which $7 * 4$ are statistical and dynamic features, and $5 * 4$ are actigraphy derived sleep parameters (see page 7 for the definitions), as shown in Table III and Table IV of the revised manuscript. This was possible as an actiwatch device (Respironics Actiwatch Spectrum Pro) was used for the data collection, which allowed 5 sleep parameters to be extracted at each intensity threshold from the signal (see reference [41]). The feature set used in the study incorporates 48 features, i.e. $(12 * 4 = 48)$ accumulating 12 features at each intensity filtering threshold. Our dataset contains 48 features (see Table IV, also [link to dataset, Dryad depository, https://datadryad.org/stash/share/l2vntnfAxMy9awK7oEbsXFtbClmJg9VvOnPk1kK5Ecl](https://datadryad.org/stash/share/l2vntnfAxMy9awK7oEbsXFtbClmJg9VvOnPk1kK5Ecl)).

In the previous publication (reference [17], Angelova et al 2020) we used $10 * 4 = 40$ features. This was because we used retrospective data, where an older actigraphy device was used for the data collection, which allowed only 3 actigraphy derived sleep parameters, TST, WASO and SWR, to be extracted from the signal at each intensity threshold.

Action: Typo corrected, $(12 * 4) = 48$, Section II, part E, first paragraph.

[III. Results, B. Classification of Chronic insomnia]

- The manuscript says that “the model performance decreases with the length (total analysed nights) (Fig. 6)”. I guess this is a typo, and the text should say “increases”, perhaps an additional comment can be added to the caption of Fig. 6 noting that the direction of the horizontal axis is opposite to the standard direction.

Response/Action: We have made the changes as suggested by the reviewer. The changes are highlighted in section III. Results, subsection B. Classification of Chronic Insomnia of the revised manuscript.

- “the optimal classification performance is achieved for 7 nights of recordings and the model classification performance drops to a random guess for a recording below 5 nights”. If you would have longer recordings, e.g., 2 weeks, would it be possible to improve still the classification? Doesn't Fig. 6 suggest that accuracy stays $>60\%$ even when using recordings of only 3 nights, and therefore well above the 50% of a random guess?

Response: We agree with the reviewer observations. In principle the model performance could increase if the number of analyzed nights are increased even after the 7 nights recording. However, we are looking at reasonable performance with minimum number of nights as it will be difficult for people to wear the device continuously for longer periods, namely the design of the study was for 7 nights as feasibility of longer period was questionable.

The findings from the study suggest (fig. 6) that an accuracy ~60% could be achieved just using 3 nights of data, which is not sufficient for accurate classification. Furthermore, as insomnia individuals could sleep well for one night and badly during another night, reducing the number of nights to < 5 nights can impede the purpose of the model as complementary to clinical assessment of insomnia (Riemann et al, 2017).

Riemann D. et al. (2017). European guideline for the diagnosis and treatment of insomnia. *J Sleep Res.* 26, 675-700. doi: 10.1111/jsr.12594.

Action: The discussion is extended to incorporate this comment in section IV. Discussions, subsection D. Conclusion and Future work of the revised manuscript.

[IV. Discussion]

- It is known that noise in a time series increases SampEn. Can you give a reference?

Response/Action: We have added a reference, (see ref [49]) in the revised version of the manuscript, section III, part A.

S. Ramdani, F. Bouchara, and J. Lagarde, "Influence of noise on the sample entropy algorithm". *Chaos: An Interdisciplinary Journal of Nonlinear Science*, 19(1), p.013123, 2009.